# The TRIple-frequency and Polarimetric radar Experiment for improving process observation of winter precipitation

José Dias Neto[1], Stefan Kneifel[1], Davide Ori[1], Silke Trömel[2], Jan Handwerker[3], Birger Bohn[4], Normen Hermes[5], Kai Mühlbauer[2], Martin Lenefer[2], and Clemens Simmer[2]

[1]Institute for Geophysics and Meteorology, University of Cologne, Cologne, Germany
[2]Institute for Geosciences and Meteorology, University of Bonn, Bonn, Germany.
[3]Institute of Meteorology and Climate Research (IMK), Karlsruhe Institute of Technology (KIT), Karlsruhe, Germany
[4]Institute of Energy and Climate Research (IEK-8), Research Centre Jülich, Jülich, Germany
[5]Institute of Bio- and Geosciences Agrosphere (IBG-3), Research Centre Jülich, Jülich, Germany

**Correspondence:** José Dias (jdiasnet@uni-koeln.de)

**Abstract.** This paper describes a two-month dataset of ground-based triple-frequency (X, Ka, and W Band) Doppler radar observations during the winter season obtained at the Jülich ObservatorY for Cloud Evolution Core Facility (JOYCE-CF), Germany. All relevant post-processing steps, such as re-gridding, offset and attenuation correction, as well as quality flagging are described. The dataset contains all necessary information required to recover data at intermediate processing steps for user-specific applications and corrections (https://doi.org/10.5281/zenodo.1341389). The large number of ice clouds included in the dataset allows for a first statistical analysis of their multi-frequency radar signatures. The reflectivity differences quantified by dual-wavelength ratios (DWR) reveal temperature regimes, where aggregation seems to be triggered. Overall, the aggregation signatures found in the triple-frequency space agree with and corroborate conclusions from previous studies. The combination of DWR with mean Doppler velocity and linear depolarization ratio enables us to distinguish signatures of rimed particles and melting snowflakes. The riming signatures in the DWR agree well with results found in previous triple-frequency studies. Close to the melting layer, however, we find very large DWR (up to $20\,\mathrm{dB}$), which have not been reported before. A combined analysis of these extreme DWR with mean Doppler velocity and linear depolarization ratio allows to separate this signature, which is most likely related to strong aggregation, from the triple-frequency characteristics of melting particles.

*Copyright statement.* TEXT

## 1 Introduction

The combined observation of clouds and precipitation at different radar frequencies is used to improve retrievals of hydrometeor properties. All methods exploit frequency-dependent hydrometeor scattering and absorption properties governed by their microphysical characteristics.

Multi-frequency retrievals are already well developed for liquid hydrometeors. For example, Hogan et al. (2005) used differential radar attenuation at 35 and 94 GHz to retrieve vertical profiles of cloud liquid water. Improved precipitation rate retrievals on a global scale are provided by the core satellite of the Global Precipitation Mission which operates a Ku-Ka Band dual-frequency radar (Hou et al., 2014). For frequencies below $\approx 10$ GHz, attenuation effects are negligible (except for

heavy rainfall or hail) and the sensitivity to non-precipitating particles, such as ice crystals, is relatively weak. Therefore, the majority of multi-frequency applications for cold clouds focuses on cloud radar systems operating at 35 GHz or 94 GHz. At these frequencies, the radars are sensitive enough to detect even sub-millimeter ice particles and cloud droplets. The sizes of large ice crystals, snowflakes, graupel, and hail are in the order of the wavelengths used to observe them (3mm, 8mm, 3cm for W, Ka, and X Band, respectively). Thus, non-Rayleigh scattering becomes important and can be used to constrain particle

size distributions, improving ice and snow water content retrievals (Matrosov, 1998; Hogan et al., 2000; Leinonen et al., 2018; Grecu et al., 2018).

Recent modeling studies (Kneifel et al., 2011b; Tyynelä and Chandrasekar, 2014; Leinonen and Moisseev, 2015; Leinonen and Szyrmer, 2015; Gergely et al., 2017) revealed that different ice particle classes like graupel, single crystals, or aggregates, can be distinguished using a combination of three radar frequencies (13, 35 and 94 GHz). Triple-frequency radar datasets from

airborne campaigns (Leinonen et al., 2012; Kulie et al., 2014) and satellites (Yin et al., 2017) confirmed distinct signatures in the triple frequency space. Ground-based triple-frequency radar measurements in combination with in-situ observations (Kneifel et al., 2015) provided the first experimental evidence for a close relation between triple-frequency signatures and the characteristic particle size, as well as the bulk density of snowfall. These early results were corroborated and refined by coinciding in-situ observations in aircraft campaigns (Chase et al., 2018) as well as by ground-based observations (Gergely

et al., 2017). A better understanding of the relations between triple-frequency signatures and snowfall properties are key for triple-frequency radar retrieval development. The connection between scattering and microphysical properties is currently addressed by novel ground-based in-situ instrumentation (Gergely et al., 2017) and triple-frequency Doppler spectra (Kneifel et al., 2016). Long-term triple-frequency datasets from various sites and radar systems are, however, needed to better understand the relations between triple-frequency signatures and clouds.

We present a first analysis of triple-frequency (X, Ka, W Band) radar observations collected over two winter months at the Jülich observatory for cloud evolution core facility, Germany (Löhnert et al., 2015). The data were corrected for known offsets and attenuation effects and re-gridded for multi-frequency studies. Section 2 describes the experimental setup and the characteristics of the X, Ka and W Band radars. Section 3 details the data processing and corrections applied. Section 4 gives a general overview of the dataset and its limitations. Section 5 presents a statistical analysis of the data with a focus on

the temperature dependency of the triple-frequency properties, signatures of riming, intense aggregation, and melting snow particles. We summarize and discuss our results in Section 6.

## 2 Measurement Site and Instruments

The "TRIple-frequency and Polarimetric radar Experiment for improving process observation of winter precipitation" (TRIPEx) was a joint field experiment of the University of Cologne, the University of Bonn, the Karlsruhe Institute of Technology (KIT), and the Jülich Research Centre (Forschungszentrum Jülich, FZJ). TRIPEx took place at the Jülich Observatory for Cloud Evolution Core Facility (JOYCE-CF $50°54'31''N, 6°24'49''E$, 111 m above mean sea level) from 11 November 2015 until 04 January 2016. The core instruments deployed during TRIPEx were three vertically pointing radars providing a triple-frequency (X, Ka, and W Band) column view of the hydrometeors aloft. All three radars were calibrated by the manufacturers before the campaign. Figure 1 sketches the positions of the instruments relative to each other and the ground surface. A large number of additional permanently installed remote sensing and in-situ observing instruments are available at the JOYCE-CF site (see Löhnert et al. (2015) for a detailed overview).

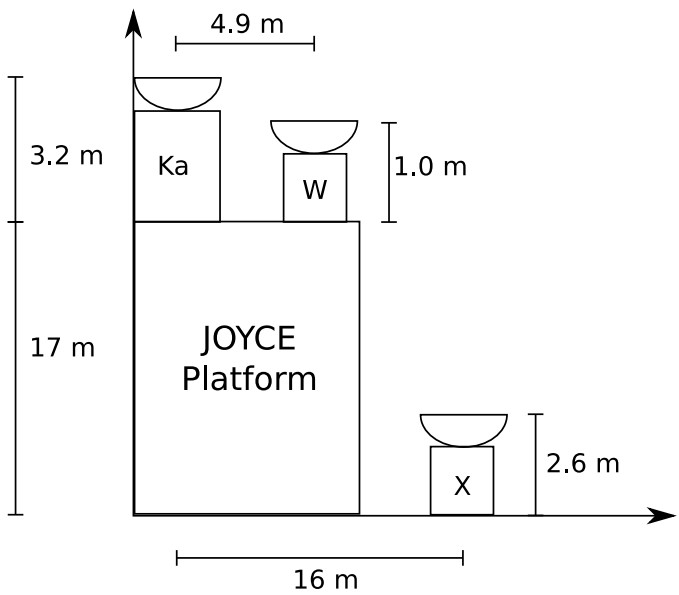

**Figure 1.** Sketch (not to scale) of the horizontal and vertical distances between the three zenith-pointing radars operated during TRIPEx. The JOYCE-CF platform with all auxiliary instruments is located on the roof of a 17 m tall building. The mobile X Band radar was placed on the ground close to the other two radars.

### 2.1 Precipitation Radar KiXPol (X Band)

KiXPol, hereafter referred to X Band, is a pulsed 9.4 GHz Doppler precipitation radar, usually integrated into the KITcube platform (Kalthoff et al., 2013). The mobile Meteor 50DX radar, manufactured by Selex ES (Gematronik), is mounted on a trailer and placed next to the JOYCE-CF building in order to position it as close as possible to the other two radars, which were installed on the JOYCE-CF roof-platform (see Figure 1). The radar operates in a simultaneous transmit and receive (STAR)

mode and is thus capable of measuring standard polarimetric variables like differential reflectivity $Z_{dr}$ and differential phase shift $\Phi_{dp}$. Linear depolarization ratio (LDR) is not provided because it requires the emission of single-polarization pulses in order to allow for independent measurements of the cross-polarized component of the returning signal. During the campaign, the X Band was set to a pulse duration of 0.3 µs; a slight oversampling was applied to achieve a radial resolution of 30 m in order to match the resolution of the other radars as close as possible (see Table 1). The X Band radar is designed for operational observations of precipitation via volume scans (series of azimuth scans at several fixed elevation angles). KiXPol was operated at JOYCE in this mode during the HOPE campaign (Xie et al., 2016; Macke et al., 2017). The standard software requires to rotate the antenna in azimuth in order to record data. Hence, we constantly rotated the antenna at zenith elevation with a slow rotation speed ($2°\mathrm{s}^{-1}$) in order to enhance the sensitivity by a longer time averaging. After each complete rotation, the radar stops the measurements for few seconds before the next scan starts, introducing thus a small measurement gap at each scan routine. Further technical specifications of X Band are listed in Table 1.

## 2.2 Cloud Radar JOYRAD-35 (Ka Band)

JOYRAD-35, hereafter referred to Ka Band, is a scanning 35.5 GHz Doppler cloud radar of the type MIRA-35 (Görsdorf et al., 2015) manufactured by Metek (Meteorologische Messtechnik GmbH), Germany. An overview of its main technical characteristics and settings used during TRIPEx is provided in Table 1. The radar transmits linearly polarized pulses at 35.5 GHz and receives simultaneously the co- and cross-polarized returns. This allows to derive LDR, which is used by the Metek processing software to filter out signals from insects and to detect the melting layer. From the measured Doppler spectra, standard radar moments such as effective reflectivity factor Ze, mean Doppler velocity (MDV) and Doppler spectral width (SW) are computed. Since March 2012, the Ka Band radar is a permanent component of JOYCE-CF (Löhnert et al., 2015), and its zenith observations are used as input for generating CloudNet products (Illingworth et al., 2007). The radar was vertically pointing most of the time because the major scientific focus during TRIPEx was to collect combined triple-frequency observations. Every 30 minutes, a sequence of Range Height Display (RHI) scans at different azimuth directions (duration $\approx$ 4 minutes) was performed in order to capture a snapshot of the spatial cloud field, and also to derive the radial component of the horizontal wind inside the cloud. The scanning data has not been processed yet; thus, the dataset described here only includes the zenith observations; the RHI scans will be included in a future release. The Ka Band radar was almost continuously operating during the TRIPEx campaign, except for a gap from 25[th] of November to 2[nd] of December 2015 due to a failure of the storage unit.

## 2.3 Cloud Radar JOYRAD-94 (W Band)

JOYRAD-94, hereafter referred to W Band, is a 94 GHz, frequency modulated continuous wave (FMCW) radar, combined with a radiometric channel at 89 GHz. The instrument is manufactured by Radiometer Physics GmbH (RPG), Germany. Unlike the X and Ka Band radar, the W Band radar is a non-polarimetric, non-scanning and non-pulsed system. W Band started measurements at JOYCE-CF in October 2015; a detailed description of the radar performance, hardware, signal processing, and calibration can be found in Küchler et al. (2017). The W Band radar has a similar beam-width, range, and temporal resolution as Ka Band (Table 1). The FMCW system allows the user to set different range resolutions for different altitude

by acting on the frequency modulation settings (chirp sequence). During TRIPEx the standard chirp sequence (Table 2) has been used. After correcting the Doppler spectra for aliasing using the method described in Küchler et al. (2017), standard radar moments such as the equivalent Ze, MDV and SW are derived.

**Table 1.** Technical specifications and settings of the three vertically pointing radars operated during TRIPEx at JOYCE-CF.

| Specifications | X Band | Ka Band | W Band |
|---|---|---|---|
| Frequency [GHz] | 9.4 | 35.5 | 94.0 |
| Pulse Repetition Frequency [kHz] | 1.2 | 5.0 | 5.3 - 12[b] |
| Number of FFT | 1200 | 512 | 512 |
| Number of Spectral Average | 1 | 20 | 8 - 18[b] |
| 3dB Beam Width [°] | 1.3 | 0.6 | 0.5 |
| Sensitivity at 5km [dBZ][a] | -10 | -39 | -33 |
| Nyquist Velocity [$\pm ms^{-1}$] | 9 | 10 | 4.2 - 9.7[b] |
| Range Resolution [m] | 30.0 | 28.8 | 16 - 34.1[b] |
| Temporal Sampling [s] | 1 | 2 | 3 |
| Lowest clutter-free range [m] | 700 | 400 | 370 |
| Radome | Yes | No | Yes |

[a] Minimum sensitivities have been derived from the reflectivity histograms shown in Fig. 8.

[b] Pulse repetition frequency, number of spectral average, Nyquist velocity and range resolution depend on the chirp definition; those values are indicated in Table 2.

**Table 2.** Main settings of the chirp sequence used during TRIPEx for the W Band radar. See Küchler et al. (2017) for a detailed description.

| | Chirp sequence | | | |
|---|---|---|---|---|
| Attributes | 1 | 2 | 3 | 4 |
| Integration Time [s] | 0.338 | 0.402 | 0.530 | 1.769 |
| Range Interval [m] | 100 - 400 | 400 - 1200 | 1200 - 3000 | 3000 - 12000 |
| Range Resolution [m] | 16.0 | 21.3 | 26.9 | 34.1 |
| Nyquist Velocity [$\pm ms^{-1}$] | 9.7 | 8.1 | 6.2 | 4.2 |
| Doppler FFT | 512 | 512 | 512 | 512 |
| Number of Spectral Average | 8 | 8 | 8 | 18 |
| Chirp Repetition Frequency [kHz] | 12.2 | 10.2 | 7.8 | 5.3 |

## 3 Data processing

The full TRIPEx dataset is structured on three processing levels. Level 0 contains the original data from X, Ka and W Band. For Level 1, the measurements are corrected for known instrument problems and sampled into a common time-height grid. At this

stage, the data can be still considered raw; further processing steps that are either dependent on radar frequency or atmospheric conditions are applied to the Level 2 dataset. These processing steps include the detection and removal of measurements affected by ground clutter, an offset correction of the radars based on independent sources, the compensation for estimated differential attenuation caused by atmospheric gases, adjustment of the DWRs by cross calibrations between the three radars, and the addition of data quality flags. These steps are meant to remove spurious multi-frequency signals, that are not produced by cloud properties. The processing is performed to the best of our knowledge, however, intermediate steps are included in the dataset in order to allow to recover the original data at any stage and apply different processing techniques. Figure 2 illustrates the work chain from Level 0 to Level 2. The following sections provide a detailed description of each step.

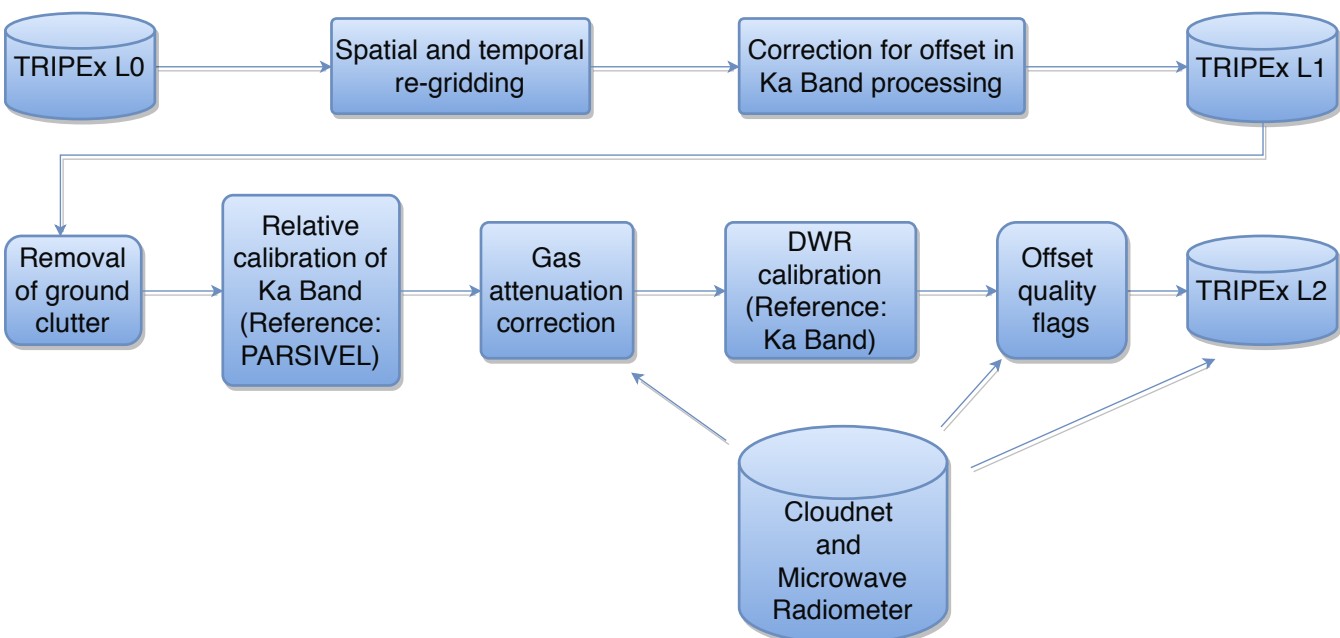

**Figure 2.** Flowchart of the TRIPEx data processing. The upper part describes the steps producing data Level 1 and the bottom part those producing data Level 2.

## 3.1 Spatio-temporal re-gridding and offset correction

Since the range and temporal resolutions of the three radars are slightly different (Table 1), the data is re-gridded on a common time and space resolution in order to allow for the calculation of dual wavelength ratios (DWRs) defined for two wavelengths $\lambda 1$ and $\lambda 2$ as

$$DWR = Ze_{\lambda 1} - Ze_{\lambda 2} \tag{1}$$

with $Ze_\lambda$ in $\mathrm{dBZ}$. The reference grid has a temporal resolution of of 4 s and a vertical resolution of 30 m, which is the resolution of W Band. The data are interpolated using a nearest-neighbour approach with the maximum data displacement limited to $\pm 17$

m in range and $\pm 2\,\mathrm{s}$ in time. This method preserves the high resolution information of the original radar observations. Limiting the interpolation displacement avoids spurious multi-frequency features that may result from non-matching radar volumes. Residual volume mismatches may occur at cloud boundaries where heterogeneities are largest. For Ka Band, two corrections are applied to the original reflectivity as suggested by the manufacturer (Matthias Bauer-Pfundstein, Metek GmbH, personal

communication). An offset of $2\,\mathrm{dB}$ is added to account for power loss caused by the finite receiver bandwidth; another $3\,\mathrm{dB}$ offset is added to correct for problems in the Digital Signal Processor used in older MIRA systems. These corrections are applied for processing of the Level 1 data.

## 3.2   Clutter removal

Following the corrections for radar offsets and re-gridding, the first step in the Level 2 processing is the removal of the range

gates affected by ground clutter. Considering the different radar installation locations (roof mount or ground surface) and antenna patterns, the clutter contamination affects each radar data differently. The thresholds for the lowest usable range gates are determined empirically and are reported in Table 1.

## 3.3   Evaluation of the Ka Band calibration with PARSIVEL disdrometer measurements

The three radars have been individually calibrated by their respective manufacturers, however, radar components might experi-

ence drifts over time which can lead to biases of several dB. The JOYCE site is equipped with a PARSIVEL optical disdrometer (Löffler-Mang and Joss, 2000) which provides the drop size distribution (DSD) with a temporal resolution of one minute. For rainfall events, the DSD can be used to calculate the associated radar reflectivity factor. In this study, the scattering properties of raindrops are calculated using the T-matrix approach (Leinonen, 2014) with a drop shape model that follows Thurai et al. (2007) and assuming drop canting angles that follow a Gaussian distribution with zero mean and $7°$ standard deviation (Huang

et al., 2008). Unfortunately, the lowest usable radar range gates are 500 - 600 m above the PARSIVEL, thus we have to assume a constant DSD over this altitude range in order to compare with the radar reflectivities. Time lags and wind shear effects raises further problems to the direct comparisons between radar measured Ze and the one calculated with PARSIVEL. For this reason, we compare only the statistical distribution of reflectivities at the lowest range gates measured over several hours with the corresponding distribution calculated at the ground level. Of course, systematic differences caused by rain evaporation,

drop breakup, or drop growth due to accretion towards the ground may affect such comparisons. However, the changes in the Ze profile are very close to the ones predicted by attenuation and constant DSD from three light rainfall cases. The reflectivity distributions from PARSIVEL and Ka Band (Figure 3) of those periods are very similar but differ by approximately $3.6\,\mathrm{dB}$ with Ka Band having the lower reflectivities. For these comparisons, periods before and after the TRIPEx campaign had to be used, because PARSIVEL had a hardware failure during the campaign. The similarity of the the results gives us indication that

this method is reliable, however, a large number of cases is still needed in other to draw a final conclusion about this method. Unfortunately, only Ka Band was available because the other two radars were not measuring during the selected rainfall events.

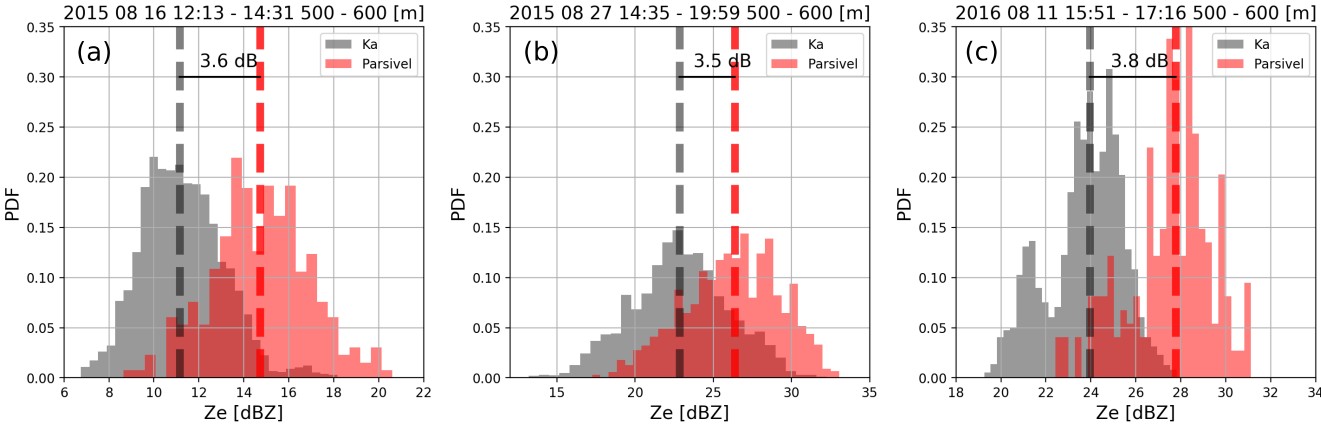

**Figure 3.** Histograms of radar reflectivities from Ka Band (gray) and results from T-matrix calculations with the rain drop size distribution provided by PARSIVEL (red) for three long-lasting stratiform rain cases before and after the TRIPEx campaign (16 August 2015 (a), 27 August 2015 (b), 11 August 2016 (c)). Ka Band reflectivities are taken from the lowest clutter-free range gates between 500 and 600 m. The vertical dashed line indicates the median of the distribution; the offset is calculated as the difference between Ka Band and T-matrix results.

### 3.4 Correction for atmospheric gas attenuation

Hydrometeors and atmospheric gases cause considerable attenuation at cloud radar frequencies. The reflectivities from X, Ka, and W Band are corrected for estimated attenuation due to atmospheric gases (Fig. 2) by means of the Passive and Active Microwave TRAnsfer model (PAMTRA) (Maahn et al., 2015). PAMTRA calculates specific attenuation due to molecular

nitrogen, oxygen and water vapor based on the gas absorption model from Rosenkranz (1993, 1998, 1999). Input parameters are the vertical profiles of atmospheric temperature, pressure and humidity provided by the CloudNet products (Illingworth et al., 2007), which are generated operationally at the JOYCE-CF site. The two-way path integrated attenuation (PIA) at the radar range gates is derived from the specific attenuation integrated along the vertical. Table 3 lists the minimum and maximum two-way attenuation values at $\approx 12$ km (height of the maximum range gate in Level 2 data) for the three radars during the entire

campaign. The highest attenuation of $\approx 2.6$ dB occurs at 94 GHz and is mainly caused by water vapour. Conversely, the 9.4 GHz maximum attenuation of $\approx 0.1$ dB is the lowest among the three radars and it is mainly produced by oxygen continuum absorption. At 35.5 GHz, attenuation is governed by both oxygen and water vapour. The maximum attenuation value found at this frequency is $\approx 0.7$ dB.

### 3.5 DWR calibration and generation of quality flags

Spurious multi-frequency signals can arise from attenuation effects due to particulate atmospheric components (e.g. liquid water, melting layer, snow), but also from instrument specific effects such as a wet radome, snow on the antenna, and remaining relative offsets due to radar mis-calibration. With this processing step, the reflectivity measurements are adjusted in order to

**Table 3.** Calculated minimum and maximum two-way path integrated attenuation (PIA) at a height of $\approx 12$ km for X, Ka, and W Band during TRIPEx.

| Frequency [GHz] | Minimum Attenuation [dB] | Maximum Attenuation [dB] |
|---|---|---|
| 9.4 | 0.077 | 0.104 |
| 35.5 | 0.365 | 0.728 |
| 94 | 0.650 | 2.675 |

take into account the cumulative effects of the aforementioned bias mechanisms at the top of the clouds. By doing so, the effects of the cloud microphysical processes on the DWR signals are recovered.

Ka Band is used as reference because of its better sensitivity level and larger dynamic range compared to the other radars (up to high altitudes) and its lower signal attenuation compared to W Band. Moreover, Ka Band is the only system not equipped with

a radome which might collect raindrops on its surface and cause additional attenuation. The signal attenuation due to antenna wetness on Ka Band is expected to be lower compared to other radars' radome attenuation because of the periodic antenna tilts during RHI scans (every 30 minutes). The processing is complemented by the generation of quality flags categorized as errors and warnings. Error flags mark data of poor quality based on the applied correction procedure, while warnings indicate the detection of potential sources of DWR offsets that have not been accounted for in the procedures described below. An

additional error flag is raised if spurious multi-frequency signals due to radar volume mismatch are suspected. A list of all the quality flags (both errors and warnings) is provided in Table 4.

The small ice particles in the upper parts of clouds are mostly Rayleigh scatterers (Kneifel et al., 2015; Hogan et al., 2000) thus, their reflectivities should not be frequency dependent (Matrosov, 1993). The reflectivity range, at which the Rayleigh approximation can be assumed, is estimated by investigating the behaviour of the observed DWRs as function of $Ze_{Ka}$. Within

15 the Rayleigh regime, the measured DWRs are expected to remain constant at a value that accounts for all the integrated differential attenuation and radar miscalibration effects. As the ice particles grow larger, the DWRs start to deviate from that constant value and this deviation affects the higher-frequency radars first. Because of that, the Rayleigh data has been isolated by means of two different reflectivity thresholds for X and W Band radars. In addition, the sensitivity of X Band is much lower; thus, a higher reflectivity threshold is accepted for the offset estimate between X and Ka Band compared to Ka and

20 W Band. For the determination of the relative offset for W Band, we found an optimal range of -30 < $Ze_{Ka}$ < -10 dBZ, and -20 < $Ze_{Ka}$ < -5 dBZ for X Band. In order to safely exclude partially melted particles, only reflectivities from at least 1 km above the 0 °C isotherm are used.

The relative offset correction is estimated for each measuring time from the data inside a moving time window of 15 minutes. The selected data are restricted to the reflectivity pairs which are within threshold values defined above. The mean value of the

25 DWR computed for these reflectivity pairs constitutes the DWR offset. The quality of this offset estimation strongly depends on the quality and quantity of the reflectivity data included in the average. Empirical analysis showed that at least 300 data points

spanning a wide reflectivity range are required in order to have acceptable sampling errors. The data that present a smaller sampling statistics are marked with an error flag.

Whenever cloud edges are included in the sampling volume and/or when the measured Ze is close to the sensitivity limits of the instruments, the correlation between the reflectivities of two radars might strongly deteriorate. In order to help the user identify these potential sources of errors, the data profiles presenting a correlation lower than 0.7 are marked with an additional error flag.

Despite the matching procedure of the different frequency radar volumes (section 3.1), mismatches are unavoidable due to the horizontal distances between the radars (Figure 1), the different radar range resolutions and beam widths (Table 1). At cloud edges and close to the melting layer, where the largest spatial cloud inhomogeneities are expected, the effects of the remaining radar volume mismatches will be maximized. The temporal DWR variability during 2-minutes moving windows is used as an indicator for potential volume mismatch; cloud regions with variances above $2 \; \mathrm{dB}^2$ are flagged accordingly.

**Table 4.** Quality flags included in the data Level 2 product (bit-coded in a 16-bit integer value). The flags indicate the reliability of the data and in relation to the quality of the relative offset estimate for X-Ka and W-Ka Band reflectivities. Note that offsets are not calculated, when the number of reflectivity pairs is below 300.

| | Bits | Criteria |
|---|---|---|
| Warning | 0-5 | Reserved for future warning flags |
| | 6 | LWP $>200 \; \mathrm{gm}^{-2}$ |
| | 7 | Rain detected by CloudNet |
| Errors | 8-12 | Reserved for future error flags |
| | 13 | Variance in time of DWR $>2 \; \mathrm{dB}^2$ |
| | 14 | Correlation of data points is poor ($<0.7$) |
| | 15 | Number of valid measurements $<300$ |

The described adjustment technique accounts for all processes that affect relative offsets of the radars in the upper and frozen part of clouds. These processes include possible frequency-dependent attenuation effects from lower levels, radar mis-calibration and radome/antenna attenuation. Since the estimated correction is applied to the entire profile, inevitably over-compensations might occur in the lower, possibly rainy parts of clouds. This limitation is necessary in order to increase the quality of the data in the ice part of the clouds, which is the main focus area of the presented study.

The lack of information about vertical hydrometeor distribution prevents reliable reflectivity corrections by differential attenuation. As a consequence of the presented DWR calibration and the fact that hydrometeor attenuation is hitting the higher frequencies more, the computed DWRs are expected to be increasingly underestimated towards the ground. A refined correction should be applied for rain and melting layer studies. Possible sources of of information about the amount and position of super-cooled liquid water could be collocated lidar or analysis of radar Doppler spectra measurements. Those data are available at JOYCE-CF but they are not included in the current dataset. However, an additional warning flag indicates periods with large

liquid water paths derived from the collocated microwave radiometer. Lastly, the occurrence of rainfall and/or a melting layer from the CloudNet classification and indicated by the precipitation gauge is marked with an additional warning flag (Table 4).

## 4 Overview of the dataset

The Level 2 of the TRIPEx dataset contains radar moments, polarimetric variables, integrated attenuation and atmospheric state
variables. The polarimetric variables are included as they are provided by the radar software and no additional processing or quality check is applied to them. $Z_{dr}$, $\phi_{dp}$ and $\rho_{hv}$ from X Band might be a useful additional source of information for melting layer studies (Zrnić et al., 1994; Baldini and Gorgucci, 2006). We are not confident about the quality of $K_{dp}$ provided by the X Band software, and therefore, this variable is not included in the dataset but can be calculated by the user. Table 5 lists all variables available in Level 2.

**Table 5.** Variables available in the TRIPEx dataset Level 2.

| Radar variables | X Band | Ka Band | W Band |
|---|:---:|:---:|:---:|
| Reflectivity [dBZ] | x | x | x |
| Mean Doppler velocity [ms$^{-1}$] | x | x | x |
| Spectral width [ms$^{-1}$] | x | x | x |
| Differential reflectivity [dB] | x | — | — |
| Differential propagation phase shift [°] | x | — | — |
| Co-polar correlation coefficient | x | — | — |
| Linear depolarization ratio [dB] | — | x | — |
| 2 way path integrated attenuation [dB] | x | x | x |
| Atmospheric variables | | CloudNet | |
| Air temperature [°C] | | x | |
| Air pressure [Pa] | | x | |
| Relative humidity [%] | | x | |

The dataset contains 47 days of measurements. For each day, table 6 lists the atmospheric conditions such as temperature at 2 m (T 2m), rain rate (RR), accumulated rain (AR), liquid water path (LWP), and integrated water vapor (IWV). The duration of four empirically classified predominant types of cloud and precipitation is provided for each day 6. The two most frequent cloud types are ice clouds (IC) with 377 hours and shallow mixed-phase clouds with 222 hours of observations. Stratiform rainfall (SR) occurred during 137 h while rain showers (SR) were only observed during 47 hours. The average rain rate (RR)
for all rainy periods over the whole period (mean rain intensity) is 0.078 mmh$^{-1}$ with a maximum instantaneous RR of 8.07 mmh$^{-1}$. DWR signatures and radar Doppler information suggest that the ice part of clouds is dominated by depositional growth and aggregation. Riming seems to occur only during a few short events. Although the dataset spans the main winter

season, no snowfall was recorded at the surface. In the following, we will demonstrate the effect of applying data quality flags and discuss remaining limitations as well as the effects of the different radar sensitivities.

**Table 6.** Characterization of the atmospheric conditions and estimated duration of cloud/precipitation events during TRIPEx. T 2m is the air temperature at 2 m from a nearby weather station. RR and AR are the rain rate and the accumulated rain measured by a Pluvio disdrometer; mean RR is calculated using all RR values larger than $0 \ \mathrm{mmh^{-1}}$. Liquid water path (LWP) and integrated water vapor (IWV) are derived from the collocated 14-channel microwave radiometer; mean LWP is calculated using all LWP values larger than $0.03 \ \mathrm{kgm^{-1}}$ in order to exclude clear-sky periods. The columns with IC, SR, RS and MP indicate the approximate duration in hours of non-precipitating ice clouds, stratiform rain, rain showers, and shallow mixed-phase clouds, respectively.

| date | T 2m [C] | RR $[\mathrm{mmh^{-1}}]$ | AR | LWP $[\mathrm{kgm^{-2}}]$ | IWV $[\mathrm{kgm^{-2}}]$ | IC | SR | RS | MP |
| yyyy.mm.dd | max / min | max / mean | [mm] | max / mean | max / mean | [h] | [h] | [h] | [h] |
|---|---|---|---|---|---|---|---|---|---|
| 2015.11.11 | 12.85 / 11.13 | 0.00 / 0.00 | 0.00 | 0.42 / 0.10 | 25.76 / 17.50 | 9 | 0 | 0 | 24 |
| 2015.11.12 | 12.81 / 10.25 | 0.00 / 0.00 | 0.00 | 0.29 / 0.07 | 20.58 / 17.34 | 18 | 0 | 0 | 18 |
| 2015.11.13 | 13.89 / 7.52 | 0.66 / 0.27 | 0.59 | 1.61 / 0.15 | 23.72 / 15.82 | 13 | 0 | 8 | 6 |
| 2015.11.14 | 10.86 / 6.46 | 0.33 / 0.12 | 0.79 | 0.38 / 0.10 | 19.34 / 12.23 | 12 | 10 | 0 | 0 |
| 2015.11.15 | 15.99 / 10.15 | 0.15 / 0.05 | 0.08 | 0.63 / 0.11 | 28.27 / 20.87 | 11 | 0 | 0 | 21 |
| 2015.11.16 | 13.74 / 11.45 | 2.16 / 0.40 | 2.16 | 2.64 / 0.15 | 28.65 / 18.99 | 4 | 4 | 3 | 12 |
| 2015.11.17 | 15.83 / 11.94 | 5.97 / 0.82 | 8.31 | 1.68 / 0.16 | 29.39 / 19.23 | 10 | 0 | 10 | 0 |
| 2015.11.18 | 14.60 / 11.41 | 8.07 / 1.88 | 4.40 | 1.65 / 0.13 | 27.71 / 15.02 | 6 | 0 | 0 | 14 |
| 2015.11.19 | 11.76 / 8.41 | 5.64 / 1.16 | 12.82 | 1.70 / 0.20 | 23.51 / 17.22 | 13 | 12 | 2 | 0 |
| 2015.11.20 | 9.45 / 4.87 | 1.08 / 0.27 | 1.02 | 0.98 / 0.13 | 19.02 / 13.63 | 10 | 3 | 0 | 6 |
| 2015.11.21 | 5.66 / 2.17 | 0.30 / 0.11 | 0.23 | 1.38 / 0.12 | 15.38 / 8.820 | 4 | 0 | 7 | 6 |
| 2015.11.22 | 5.33 / -0.09 | 7.35 / 3.80 | 2.54 | 0.84 / 0.07 | 11.11 / 8.17 | 4 | 0 | 5 | 2 |
| 2015.11.23 | 5.32 / -0.42 | 0.00 / 0.00 | 0.00 | 0.52 / 0.08 | 9.81 / 7.83 | 7 | 0 | 0 | 2 |
| 2015.11.24 | 4.51 / 0.19 | 1.26 / 0.28 | 1.30 | 0.53 / 0.17 | 16.71 / 12.57 | 10 | 12 | 0 | 0 |
| 2015.12.03 | 11.90 / 6.63 | 0.00 / 0.00 | 0.00 | 0.03 / 0.03 | 15.38 / 13.59 | 10 | 0 | 0 | 5 |
| 2015.12.04 | 11.39 / 5.87 | 2.67 / 0.56 | 3.38 | 0.57 / 0.21 | 24.09 / 10.98 | 4 | 4 | 0 | 2 |
| 2015.12.05 | 10.20 / 4.47 | 0.00 / 0.00 | 0.00 | — | 9.77 / 7.19 | 16 | 0 | 0 | 0 |
| 2015.12.06 | 12.86 / 3.34 | 0.00 / 0.00 | 0.00 | 0.39 / 0.11 | 24.14 / 15.63 | 2 | 0 | 0 | 12 |
| 2015.12.07 | 14.53 / 8.74 | 0.03 / 0.03 | 0.00 | 0.51 / 0.13 | 24.31 / 18.81 | 9 | 0 | 4 | 8 |
| 2015.12.08 | 14.66 / 7.92 | 2.67 / 0.84 | 4.06 | 0.84 / 0.18 | 23.01 / 14.67 | 2 | 5 | 0 | 0 |
| 2015.12.09 | 9.34 / 2.20 | 0.06 / 0.03 | 0.04 | 0.48 / 0.08 | 18.89 / 8.96 | 0 | 4 | 0 | 1 |
| 2015.12.10 | 8.81 / 0.77 | 0.00 / 0.00 | 0.00 | — | 11.86 / 6.49 | 7 | 0 | 0 | 0 |
| 2015.12.11 | 8.61 / 4.77 | 2.16 / 0.57 | 9.34 | 0.41 / 0.17 | 19.81 / 16.18 | 2 | 20 | 0 | 0 |
| 2015.12.12 | 10.42 / 4.7 | 0.03 / 0.03 | 0.02 | 0.36 / 0.09 | 21.10 / 15.73 | 16 | 0 | 0 | 0 |
| | | | | continue next page | | | | | |

| date | T 2m [C] | RR [mmh$^{-1}$] | AR | LWP [kgm$^{-2}$] | IWV [kgm$^{-2}$] | IC | SR | RS | MP |
|---|---|---|---|---|---|---|---|---|---|
| yyyy.mm.dd | max / min | max / mean | [mm] | max / mean | max / mean | [h] | [h] | [h] | [h] |
| 2015.12.13 | 10.08 / 6.18 | 3.09 / 0.37 | 5.50 | 1.07 / 0.38 | 22.73 / 19.10 | 7 | 0 | 0 | 8 |
| 2015.12.14 | 9.24 / 3.36 | 0.03 / 0.03 | 0.02 | 0.17 / 0.08 | 16.00 / 12.95 | 6 | 0 | 0 | 0 |
| 2015.12.15 | 10.3 / 3.89 | 0.39 / 0.16 | 0.16 | 0.57 / 0.15 | 23.55 / 17.51 | 12 | 2 | 3 | 0 |
| 2015.12.16 | 13.04 / 8.90 | 2.49 / 0.39 | 6.02 | — | — | 0 | 10 | 0 | 7 |
| 2015.12.17 | 16.28 / 12.53 | 3.60 / 0.48 | 0.72 | 1.12 / 0.15 | 25.61 / 20.01 | 8 | 0 | 0 | 6 |
| 2015.12.18 | 13.11 / 8.74 | 0.27 / 0.17 | 0.08 | 0.71 / 0.12 | 26.64 / 16.45 | 10 | 0 | 1 | 2 |
| 2015.12.19 | 13.21 / 9.93 | 0.00 / 0.00 | 0.00 | 0.27 / 0.09 | 25.11 / 22.70 | 8 | 0 | 0 | 0 |
| 2015.12.20 | 13.22 / 11.31 | 0.00 / 0.00 | 0.00 | 0.44 / 0.10 | 23.15 / 20.99 | 22 | 1 | 0 | 0 |
| 2015.12.21 | 12.17 / 9.52 | 0.72 / 0.18 | 0.45 | 0.84 / 0.13 | 23.52 / 14.49 | 3 | 3 | 1 | 6 |
| 2015.12.22 | 14.75 / 10.41 | 2.19 / 0.41 | 1.45 | 0.61 / 0.08 | 26.53 / 22.00 | 16 | 2 | 0 | 8 |
| 2015.12.23 | 13.00 / 4.38 | 0.45 / 0.21 | 0.42 | 0.23 / 0.07 | 14.21 / 11.24 | 4 | 0 | 0 | 8 |
| 2015.12.24 | 14.51 / 4.38 | 5.34 / 0.68 | 1.82 | 1.14 / 0.11 | 22.91 / 15.40 | 6 | 0 | 1 | 3 |
| 2015.12.25 | 13.35 / 7.78 | 3.27 / 0.81 | 4.72 | 0.60 / 0.13 | 24.76 / 18.32 | 15 | 8 | 0 | 4 |
| 2015.12.26 | 15.78 / 7.17 | 0.00 / 0.00 | 0.00 | 0.20 / 0.08 | 22.51 / 17.55 | 4 | 0 | 0 | 4 |
| 2015.12.27 | 14.40 / 6.13 | 0.00 / 0.00 | 0.00 | — | 18.71 / 14.20 | 12 | 0 | 0 | 0 |
| 2015.12.28 | 11.07 / 5.12 | 0.00 / 0.00 | 0.00 | — | 9.56 / 8.57 | 11 | 0 | 0 | 0 |
| 2015.12.29 | 11.87 / 4.35 | 0.00 / 0.00 | 0.00 | 0.34 / 0.08 | 19.78 / 13.80 | 2 | 3 | 0 | 0 |
| 2015.12.30 | 9.40 / 3.77 | 0.00 / 0.00 | 0.00 | 0.05 / 0.04 | 17.80 / 10.93 | 3 | 0 | 0 | 0 |
| 2015.12.31 | 10.31 / 3.53 | 0.69 / 0.20 | 0.47 | 1.01 / 0.22 | 24.39 / 11.82 | 4 | 3 | 2 | 0 |
| 2016.01.01 | 8.45 / 3.46 | 0.30 / 0.13 | 0.10 | 0.83 / 0.13 | 15.42 / 9.85 | 13 | 0 | 0 | 6 |
| 2016.01.02 | 5.94 / 4.11 | 2.88 / 0.72 | 4.69 | 0.42 / 0.14 | 17.80 / 12.89 | 6 | 7 | 0 | 8 |
| 2016.01.03 | 8.29 / 4.84 | 1.86 / 0.44 | 2.95 | 0.93 / 0.23 | 19.85 / 14.45 | 6 | 14 | 0 | 4 |
| 2016.01.04 | 7.74 / 3.66 | 3.57 / 0.81 | 7.06 | — | — | 0 | 10 | 0 | 9 |
| Total |  |  |  |  |  | 377 | 137 | 47 | 222 |

## 4.1 Effects of data filtering based on quality flags

The effects of data filtering on DWR$_{XKa}$ and DWR$_{KaW}$ is demonstrated for clouds observed on 20.11.2015 in Figures 4 and 5. In order to give a better visual impression of these effects, the filtering steps are applied sequentially and cumulatively. Panels a-c of Figure 4 show the unfiltered Level 2 data. The time-height plot (panels a and b in Figure 4) reveal a stratiform cloud passing over the site from 01:00 to 17:00 UTC followed by a series of low-level, shallow, most likely mixed-phase clouds. The short periodic gaps result from interruptions of zenith observations caused by range-height indicator (RHI) scans of Ka Band, and the large gap in DWR$_{KaW}$ between 09:00 and 10:00 UTC is caused from missing W Band observations. The -15 °C isotherm (dashed line in the time-height plots) separates DWRs around 0 dB for temperatures below -15 °C from rapid increases with reflectivity for higher temperatures.

Panel c in Figure 4 displays a scatter density plot of DWR$_{XKa}$ versus DWR$_{KaW}$ (hereafter called triple-frequency plot). The position in the triple-frequency plot is mainly driven by the respective hydrometeors bulk density $\rho$ and their mean volume diameter $D_0$ (Kneifel et al., 2015). This plot allows to discriminate between the two processes: rimed particles follow the flat curve (low DWR$_{XKa}$) due to their higher density, while aggregated particles give rise to a bending-up signature (increase in DWR$_{XKa}$ while DWR$_{KaW}$ saturates or even decreases) due to their lower density, which is nicely shown in Figure 4, Panel c.

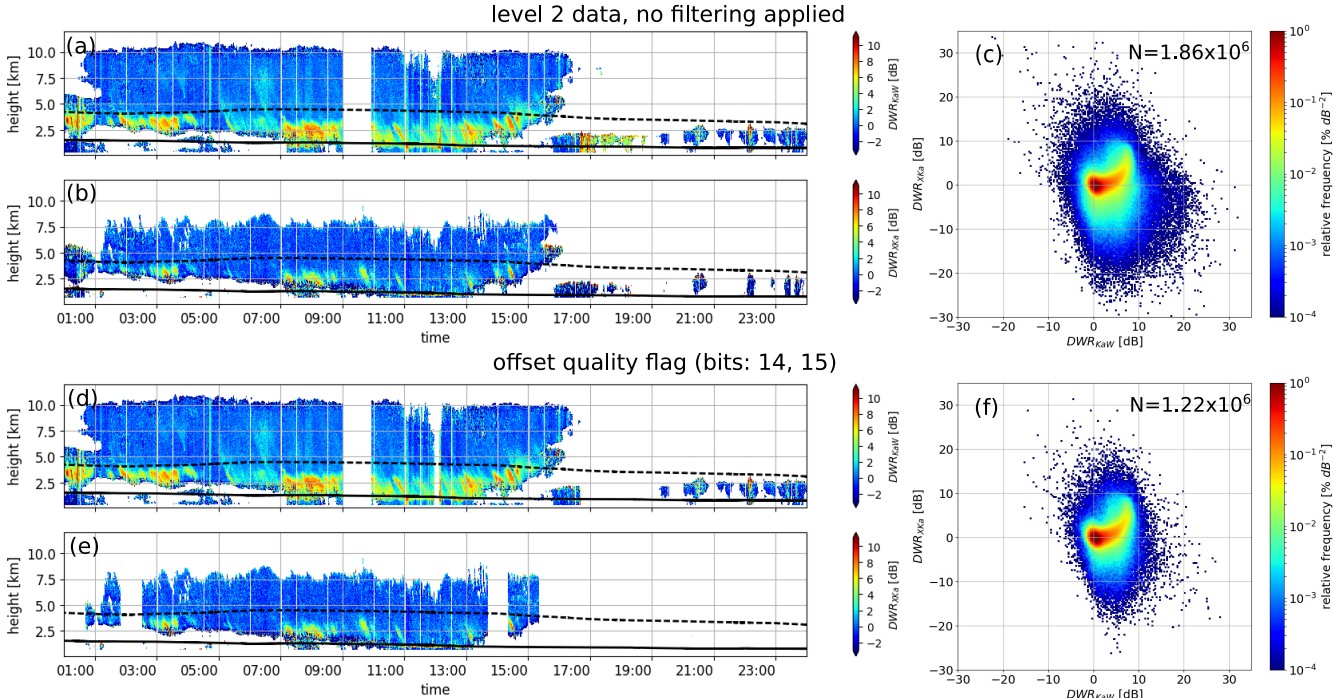

**Figure 4.** Time-height plots of DWR$_{KaW}$ (Panel a) and DWR$_{XKa}$ (Panel b) using the Level 2 data of 20.11.2015 without applying any filtering. The continuous line and dashed line are the 0 and -15 $^{\circ}$C isotherms (provided by the CloudNet products), respectively. The triple-frequency signatures for the ice part of the clouds are shown on the right (Panel c). Panels d-f show the remaining data after applying the offset quality flags and the restriction to data pairs with sufficient correlation. N in Panels c and f indicates the respective number of data pairs in the ice part of the clouds. Note the log-scale on the colorbars in c and f.

A large number of points in Panel c, Figure 4 populate areas which are unrealistic from a microphysical point, such as negative DWRs. Some of those originate from time periods when the offset cannot be calculated properly or when the correlation between the three radars is poor. Panels d and e in Figure 4 show the results after removing those points (bits 14 and 15 in the quality flag, see Table 4), an effect best visible between 17:00 and 20:00 for DWR$_{KaW}$ and between 17:00 and 23:00 for DWR$_{XKa}$. The triple-frequency plot (Panel f in Figure 4) shows a strong reduction of outliers, when compared to the unfiltered triple-frequency plot (Panel c in Figure 4).

Despite the data filtering described in the previous paragraph, the scatter around the main signature is still large. Panels a and b in Figure 5 show the time-height plots after removing observations flagged with the DWR 2-minute temporal variance

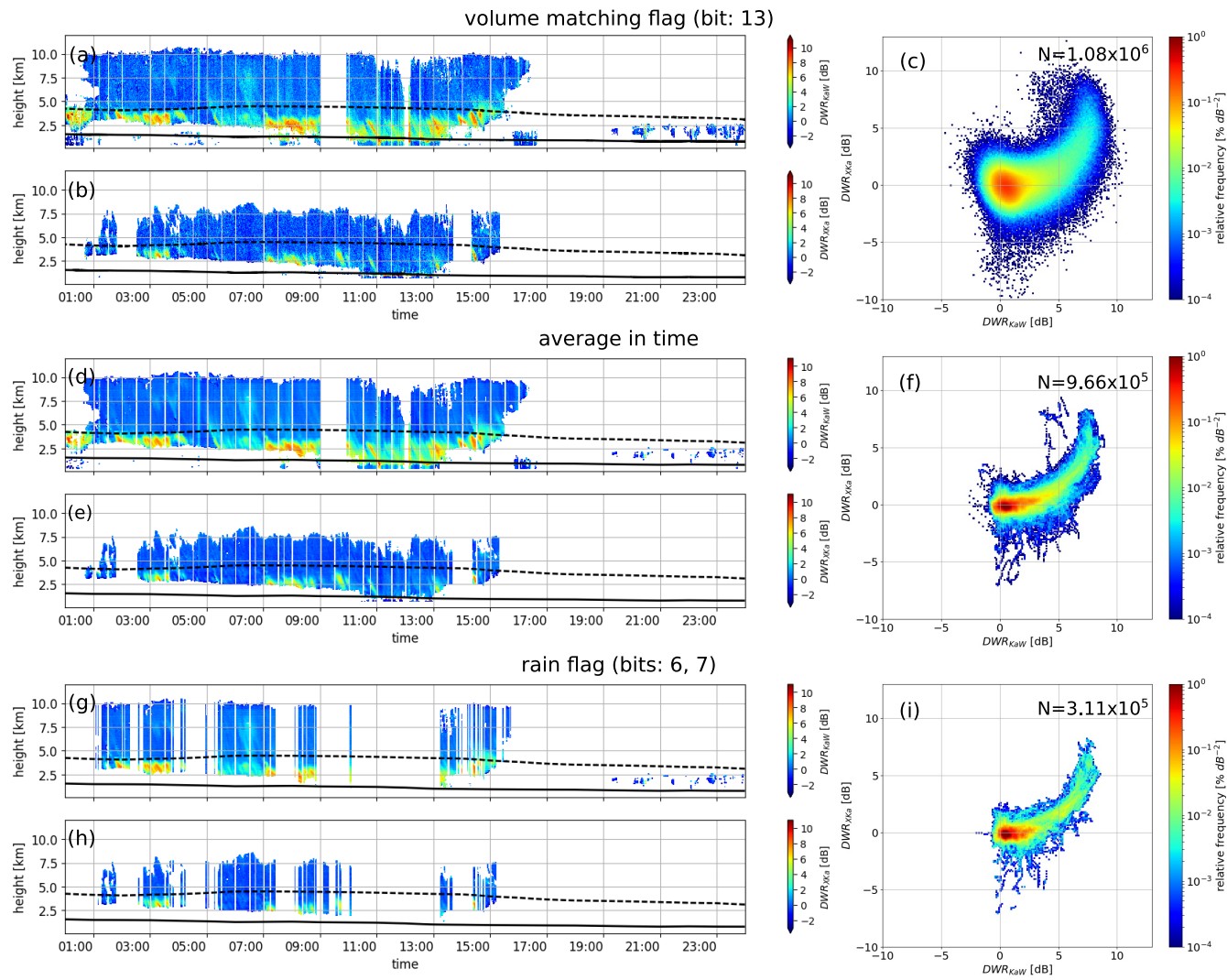

**Figure 5.** Same as Figure 4, but here the effects of cumulative data filtering subject to different quality flags and averaging is illustrated. Panels a-c display the effect of filtering based on the DWR variance in time, which removes areas potentially affected by poor radar volume matching. The effect of an additional temporal averaging over 3 minutes is shown in Panels e-f. The effect of the removal of time periods with rain as identified by CloudNet or large liquid water paths measured by the nearby microwave radiometer are displayed in Panels g-i. Note the log-scale on the colorbars in c, f and i.

flag (bit 13 in the quality flag, see Table 4). This filtering step removes most of the outliers from the aggregation signature in the triple-frequency plot (Panel c in Figure 5). It is worth noting that the removal of such data reduces the scatter in the triple-frequency space, but might also remove interesting measurements from regions with strong reflectivity gradients. An additional 3-minutes running-window averaging of the reflectivities keeps the most stable signatures (Panels d and e in Figure 5), further removes scatter, and thus accentuates the aggregation signature in triple-frequency plot (Panel f in Figure 5). The averaged reflectivities, calculated in this procedure, are not included in the TRIPEx dataset because it would not be possible to retrieve the original data. The last two quality flags (bits 7 and 6, see Table 4) mark data acquired during rainfall according to the CloudNet product and times with total liquid water path larger than $200\,\mathrm{gm^{-2}}$ as estimated by the microwave radiometer. The latter filtering significantly reduces the amount of usable data (Panel g and h in Figure 5), but preserves the main aggregation signature surprisingly well (Panel i in Figure 5).

## 4.2 Limitations of the current dataset

Despite the filtering steps discussed in Section 3.5, some limitations remain. As an example, on 23.11.2015 between 16:00 and 23:00 UTC we observe enhanced values of $Ze_X$ (-20 up to 10 dBZ) (Figure 6, Panel a), while $Ze_{Ka}$ and $Ze_W$ remain very low. The mean Doppler velocity of that structure is very small (MDV between 0 and $0.5\,\mathrm{ms^{-1}}$) and is associated with strongly enhanced LDR from Ka Band (Figure 6, Panel b). Large $Z_{dr}$ values are observed by the near-by weather polarimetric X Band radars JuXPol and BoXPol (see Diederich et al. (2015) for a detailed characterization of the radars), that were performing RHI scans over the TRIPEx site at that time. The most likely explanation based on the polarimetric signature and the fall velocity are fall streaks of chaff deployed by military aircraft during a training session. We recommend to avoid this period for cloud microphysical studies.

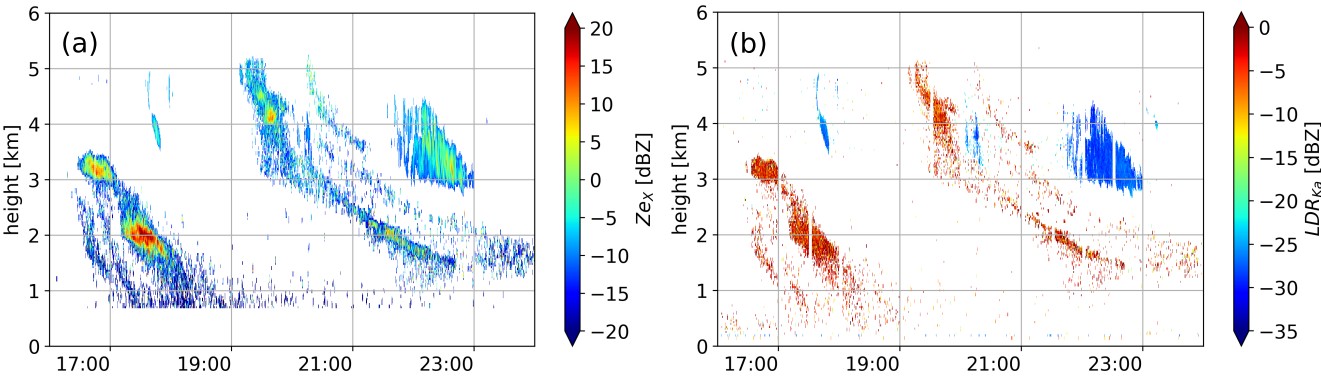

**Figure 6.** Time-height plots of the $Ze_X$ and $LDR_{Ka}$ of 23.11.2015 between 16:00 and 23:59. The region where the LDR is ≈ -5 dB is most probably a results of chaff. The Ka Band software applies a filtering for non-meteorological targets which removes most of the chaff; only the filtered Ka Band data are included in the TRIPEx dataset. Note, that no such filtering is applied to the X Band and W Band data.

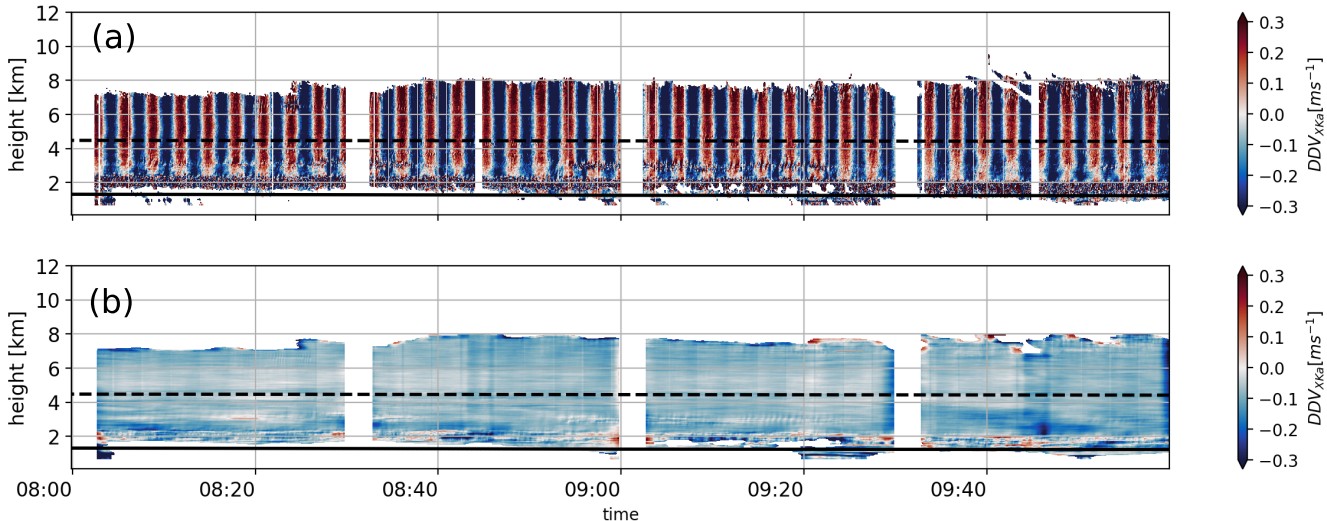

**Figure 7.** Time-height plots of the dual mean Doppler velocity using the Level 2 data of 20.11.2015. The dashed line and the continuous line are the -15 and 0 $^\circ$C isotherms, respectively. Panel a shows the DDV$_{\mathrm{XKa}}$ using the original data from Level 2. Panel b shows the DDV$_{\mathrm{XKa}}$ after applying a 3 minutes moving average.

As described in Section 2.1, X Band was operated vertically pointing while rotating the antenna. Figure 7 illustrates effects related to imperfect vertical antenna pointing. When looking at differences between vertical Doppler velocities observed from low frequency and high frequency radars (Dual Doppler velocity, DDV), increases are expected in presence of large scatterers (Matrosov, 2011; Kneifel et al., 2016). Large particles, which usually also have greater terminal velocities, give a lower reflec-
tivity signal at high frequencies due to non-Rayleigh scattering. This effect also leads to a lower MDV (MDV$_{\mathrm{X}}$ > MDV$_{\mathrm{Ka}}$ > MDV$_{\mathrm{W}}$). Since the ice particles in the uppermost part of the clouds are expected to be Rayleigh scatterers, the DDV should be zero. However, DDV$_{\mathrm{XKa}}$ (Panel a of Figure 7) shows a periodic variation along the entire vertical range with the period matching the X Band scan duration of 3 minutes. Obviously, a non-perfect zenith pointing of the X Band antenna introduces these periodic shifts in the mean Doppler velocity due to the contamination of the vertical Doppler signal by the horizontal
wind component. A temporal average over 3 minutes minimizes the standard deviation of DDV$_{\mathrm{XKa}}$ relative to other averaging window sizes (Panel b of Figure 7). Note, that the averaged data is not included in the Level 2 data product, because the optimal averaging window might depend on the prevailing atmospheric, height-dependent wind conditions, and original data cannot be recovered after averaging. We can also not completely rule out a slight mispointing of the other two radars, because their DDVs sometimes show deviations especially in regions with strong horizontal winds with maximum DDVs. However, these
DDVs are found to be below $0.4\,\mathrm{ms}^{-1}$. An ad-hoc estimate of the related relative radar mispointing of the two radars using the horizontal wind information from radiosondes for a few extreme cases suggests a potential mismatching of 0.5 $^\circ$. A correction of the shift requires reliable horizontal wind profiles, which will be investigated in more detail in the future.

### 4.3 Radar sensitivity

Figure 8 shows the distribution of reflectivity values measured by the three radars during the entire campaign filtered with the error flags (bits 13, 14 and 15 in Table 4), and stratified by height above the site. As already mentioned, Ka Band and W Band show higher sensitivities compared to X Band up to high altitudes. Ka Band (Panel b in Figure 6) exhibits the largest dynamic range (Panels a and c in Figure 6). The step-like shape of the lowest altitude reflectivities from W Band is caused by different chirp settings (Table 2). A polynomial fit to the minimum retrieved linear reflectivities ($Ze_{lin}$ in units of $mm^6/m^3$) as a function of altitude $z$ (units of m),

$$Ze_{lin}(z) = a \cdot z^b \tag{2}$$

results for X and Ka Band in the expected nearly quadratic decrease with range (Table 6). The slower decrease (smaller exponent) for W Band results from the altitude-dependent sensitivity associated with the height-varying chirp settings.

The melting layer was mostly observed at altitudes between 1 and 2 km where it causes a sharper increase in the reflectivity distribution and the largest values measured for the X Band reflectivities. The X band Ze distribution shows an enhancement of the largest recorded values at 2 km from ≈30 dBZ to ≈40 dBZ. The X Band sensitivity limitations did not allow to observe signals above 7 km with reflectivities below -10 dBZ, however, dual-wavelength studies of clouds in this region are still possible with the W Band and Ka Band included in the Level 2 data. Nonetheless, ice aggregation and riming, which are most relevant for triple-frequency studies, usually occur at lower levels and larger reflectivities where all three radars provide sufficient sensitivity.

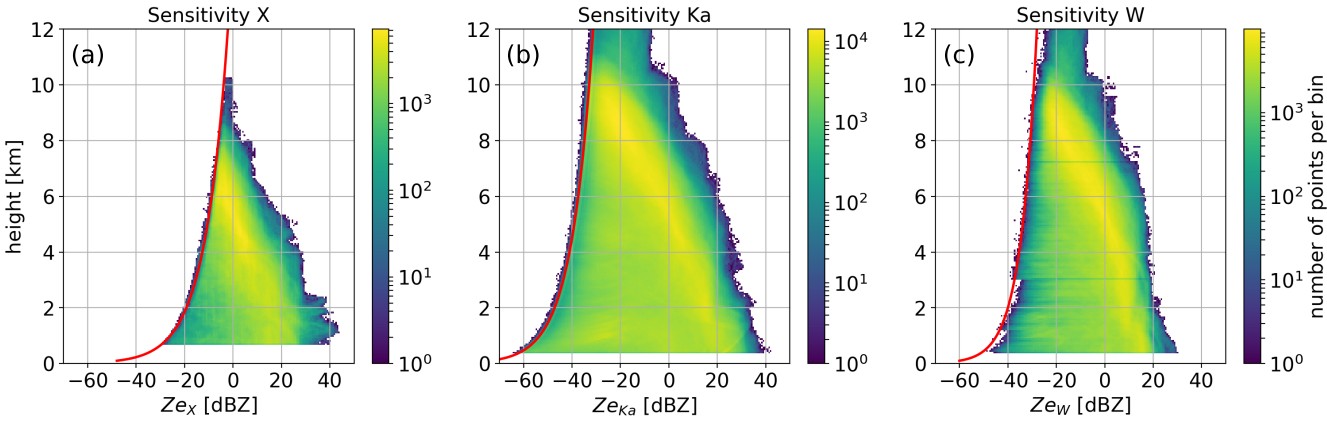

**Figure 8.** Histograms of reflectivities from the entire TRIPEx campaign Level 2 data for each radar. The red curve is the profile of the minimum retrieved reflectivity (Eq. 2). Panels a, b and c show the histograms for X, Ka and W Band, respectively; all error flags (see Tab. 4) were applied to filter the data. Note the log-scale on the colorbars.

**Table 6.** Coefficients $a$ and $b$ for the sensitivity fit ( Eq. 2) obtained for X, Ka and W Band. The coefficients were calculated using the Level 2 dataset with a filtering according to the error flags applied (see Table 4).

| Radar | $a$ | $b$ |
|---|---|---|
| X Band | $6.25 \cdot 10^{-10}$ | 2.19 |
| Ka Band | $3.41 \cdot 10^{-12}$ | 2.04 |
| W Band | $8.36 \cdot 10^{-10}$ | 1.53 |

## 5 Triple-frequency characteristics of ice and snow clouds

Longer time series of observations are required in order to reliably estimate the occurrence probabilities of process signatures in the triple-frequency space. Those statistics might be useful for the development of microphysical retrievals and to constrain snow particle scattering models. Currently available datasets are restricted to short time periods or specific cases. Kulie et al. (2014) and Leinonen et al. (2012) used observations from airborne Ku, Ka and W Band radars data collected during the Wakasa Bay campaign (Lobl et al., 2007) to evaluate aggregate and spheroidal snowflake models. Their $DWR_{KaW}$ and $DWR_{KuKa}$ values reach up to 10 dB and 8 dB, respectively. Although their data are rather noisy due to volume mismatch and attenuation effects, they were the first observations which confirmed triple-frequency signatures predicted by complex aggregate scattering models (Kneifel et al., 2011a). The first triple-frequency signatures from ground-based radars (S, Ka, W Band) were presented by Stein et al. (2015) for two case studies. Similar to the Wakasa Bay studies, they found deviation from predictions based on simpler spheroidal-based scattering models, but their aggregates showed a $DWR_{KaW}$ saturation around 8 dB and not the 'hook' or 'bending back' feature found in the previous studies. They attributed this behaviour to a snow aggregate fractal dimension of two. Kneifel et al. (2015) combined triple-frequency ground-based radar (X, Ka and W Band) with in-situ observations, and analyzed three cases characterized by falling snow particles with different degrees of riming. For low density aggregates their $DWR_{KaW}$ did also not exceed the 8 dB limit reported by previous studies, but exhibited a strong bending back feature (i.e., reduction of $DWR_{KaW}$ for larger particles) with large $DWR_{XKa}$ up to 15 dB. During riming periods, the triple-frequency signatures showed a distinctly different behavior: $DWR_{KaW}$ increases up to 10 dB, while $DWR_{XKa}$ remains constant or slowly increases up to 3 dB, which appears in triple-frequency plots as an almost horizontal line.

The TRIPEx dataset is, to the best of our knowledge, one of the longest, quality-controlled triple-frequency datasets currently available, which allows for reliable estimations of the occurrence of several triple-frequency signatures in mid-latitude winter clouds. In the following sections, we use the Level 2 data filtered only with the errors quality flag (see Table 4) to analyze the temperature dependence of the triple-frequency signatures and signatures of riming and melting snow particles. The extension of the filtering to the warning flags would remove all melting layer cases and/or observations with larger amounts of super-cooled liquid water, which portray particularly interesting signatures of partially melted or rimed particles.

## 5.1 Temperature dependence of triple-frequency signatures

The relatively large dataset allows us to stratify the occurrence probability of $DWR_{KaW}$ (Panel a in Figure 9) and $DWR_{XKa}$ (Panel b in Figure 9) according to air temperature, which results in four main regimes. The regime where the temperature is smaller than -20 °C exhibits small DWR values, mostly below 3 dB.

Between -20 and -10 °C, we find a widening of the distribution to higher values in both DWRs. This DWR increase becomes very rapid at temperatures warmer than -15 °C, which suggests an increasing number of larger aggregates caused by stronger aggregation due to preferential growth of dendritic particles in the -20 to -10 °C temperature range (Kobayashi, 1957; Pruppacher and Klett, 1997). Dendrites are well known to favor snow aggregation due to their branched crystal structure. In accordance with previous studies, $DWR_{KaW}$ saturates around 7 dB at -10 °C with only a small fraction reaching up to 10 dB. $DWR_{XKa}$ approaches maximum values of 5 to 8 dB, however, the occurrence probability of enhanced $DWR_{XKa}$ is smaller compared to those found for $DWR_{KaW}$. This is an expected behavior since early aggregation is likely to first enhance the $DWR_{KaW}$ because particles growth early affects the high frequencies which first transition out of the Rayleigh regime. Thus W Band radar is the first influenced by this transition which enhances $DWR_{KaW}$.

At temperatures between -10 and 0 °C, the distribution of $DWR_{KaW}$ remains almost constant with the exception of a small peak with higher values around -5 °C and a widening of the DWR distributions towards negative values. The latter effect might relate to two causes. The first is the DWR calibration (Sec 3.5), derived for the upper part of the clouds (ice part), which, when applied to the entire profile, leads to the overestimation of $Ze_W$. The second possible motivation is the radar volume mismatch, which becomes worse for observations closer to the radars due to reduced overlap of the radar beams.

Interestingly, $DWR_{XKa}$ grows continuously up to 12 dB for temperatures warmer than -5 °C, which is in line with intensified aggregation of the snow particles towards lower heights. The very large $DWR_{XKa}$ in this regime can be explained by increasing particle stickiness when approaching the 0 °C level. In the fourth regime between 0 °C and the LDR maximum, $DWR_{KaW}$ tends to further increase while $DWR_{XKa}$ remains constant or even decreases. $DWR_{KaW}$ reaches values up to 10 dB while $DWR_{XKa}$ attains values up to 15 dB which could be produced by persistent aggregation.

Figure 10 shows the triple-frequency plots for the temperature ranges -20 < T < -10 °C (panel a) and -10 < T < -1 °C (panel b). Between -20 and -10 °C (panel a), we find the typical bending signature in the triple-frequency space, saturating at about a $DWR_{KaW}$ of 8 dB, similar to Stein et al. (2015). This temperature regime includes the Dendritic Growth Zone (DGZ), which is usually defined by cloud chamber experiments in the range of temperatures -17 to -12 °C (Kobayashi, 1957; Yamashtta et al., 1985; Takahashi, 2014). It is worth reminding that the temperature information, included in the TRIPEx dataset, has not been obtained from a direct measurement, but it has been taken from CloudNet. Consequently, it is not surprising that the growth regimes that we have identified using the signatures observed in the DWR profiles do not perfectly correspond in temperature to the ones determined in cloud chamber experiments.

Although we combine observations from different clouds, the variability of the triple-frequency signatures is relatively small. For warmer temperatures (-10 to -1 °C, Panel b), needle aggregates are likely to be generated and ice particles start to become more sticky leading to a more pronounced bending feature. For $DWR_{XKa}$ reaching up to 12 dB, also the hook (or bending back)

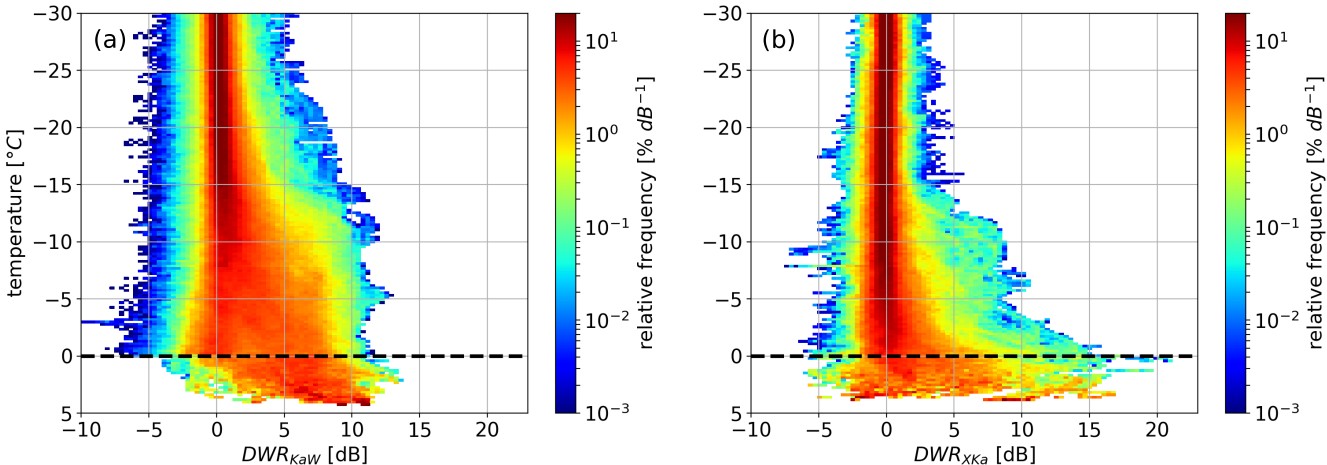

**Figure 9.** Two dimensional histograms (Contoured Frequency by Altitude Diagram (CFAD), see Yuter and Houze (1995) for more details) of DWR against air temperature for the entire TRIPEx dataset. The dashed line indicates the $0\,°C$ isotherm. The data below the dashed line are collected only from the cases where a melting layer is observed. The DWRs were filtered using the error flags and averaged in time using a 3 minutes moving window. Panels a and b show $DWR_{KaW}$ and $DWR_{XKa}$, respectively. Note the log-scale of the colorbars.

signature (Kneifel et al., 2015) becomes visible for parts of the dataset ($DWR_{XKa}$ decreases while $DWR_{KaW}$ is still increasing). This panel also reveals a secondary mode with $DWR_{XKa}$ below $3\,dB$ and $DWR_{KaW}$ reaching up to $12\,dB$. Following Kneifel et al. (2015), this mode could hint at rimed particles, which are still too small to enhance $DWR_{XKa}$, but due to their increased density and hence larger refractive index, the $DWR_{KaW}$ increases. We will investigate this feature in more detail in the next
subsection.

The dataset contains particularly large DWR signatures close to $0\,°C$ and at higher temperatures, which are probably caused by melting snowflakes or simply by enhanced aggregation. To further investigate this signature we generated the triple-frequency plot for the data between the $0\,°C$ and the height of the LDR maximum (Figure 11), which we consider as proxy for the center of the melting layer (Le and Chandrasekar, 2013). In this region, $DWR_{XKa}$ reaches maximum values up to $20\,dB$
already at low $DWR_{KaW}$. Overall, the data points are much more scattered than those in the colder temperature regions. This larger variability might result from effects of the radar volume mismatch caused by strong vertical gradients near the melting layer. Another possible explanation is the much lower amount of data. Latent heat release by melting increases turbulent motion, which might further enhance the detrimental effects of volume mismatch. We need to be careful in interpreting these features as triple-frequency signatures of the melting layer, because the temperature information is based on CloudNet products
taken from ECMWF analyses which cannot be expected to represent small scale variations of the $0\,°C$ isotherm. Moreover, melting can be delayed depending on the profiles of temperature and humidity, and on the density and size of the particles themselves (Matsuo and Sasyo, 1981; Rasmussen and Pruppacher, 1982). A sagging of the melting layer has been repeatedly observed with the scanning polarimetric X Band radar in Bonn (BoXPol, also part of JOYCE-CF) for dominant riming pro-

cesses (Xie et al., 2016; Trömel et al., 2018). Rimed particles fall with higher terminal velocities and consequently take more time to melt. In the following subsection, we will use LDR and the mean Doppler velocity to better separate non-melted from melted snow particles.

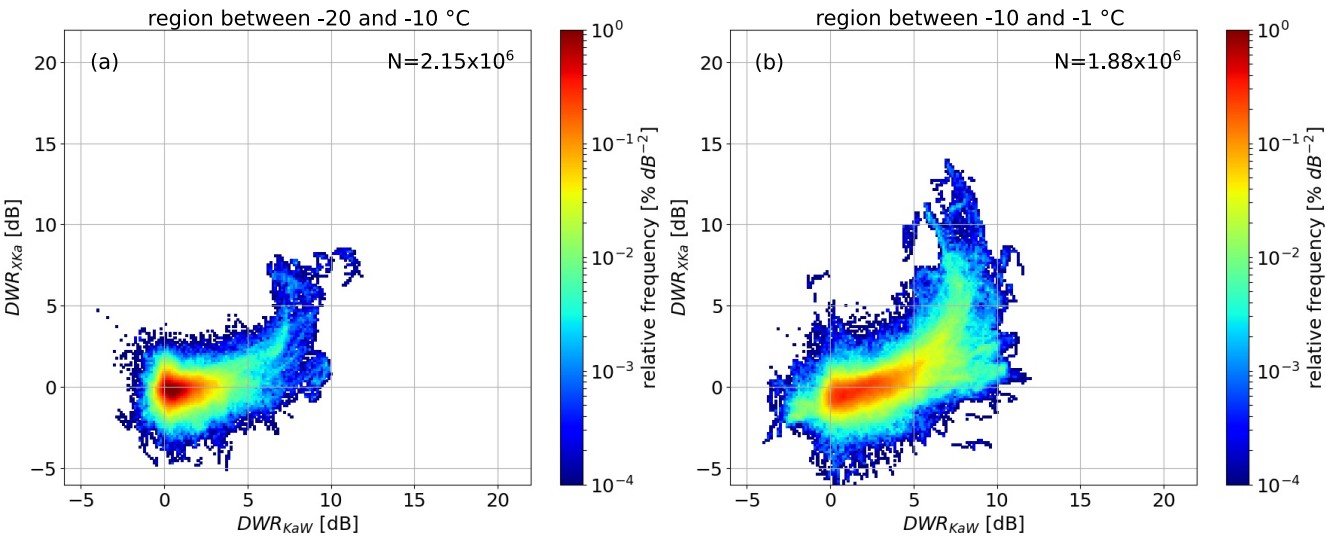

**Figure 10.** Two dimensional histogram of the triple frequency signatures for different temperature regions normalized by the total number of points N. The color shows the relative frequency. Panel a is for temperatures lower than -17 °C; Panel b shows the region between -17 °C and -7 °C; Panel c illustrates data points lying between -7 °C and -1 °C. Note the log-scale on the colorbars.

## 5.2 Signatures of riming and melting snow particles

During riming, super-cooled liquid water droplets freeze onto the ice particles. This strongly increases the particle mass while its size grows more slowly, especially during the onset of riming. Since the terminal velocity is mainly governed by the relation between particle mass (gravitational force) and its cross section perpendicular to the air stream (drag force), its terminal velocity observed by the mean Doppler velocity (MDV) increases due to riming (Mosimann, 1995). MDVs above $1.5 \text{ ms}^{-1}$ can be used as a simple indicator of rimed particles as long as vertical air velocities are small (Mosimann, 1995). About $1\%$ of

triple-frequency data in the temperature range between -20 and -1 °C have a MDV above $1.5 \text{ ms}^{-1}$ (Figure 12). Interestingly, we find one mode very similar to a sloped line found for rimed particles in Kneifel et al. (2015), which coincides with large MDVs up to $2.4 \text{ ms}^{-1}$ and DWR$_{\text{KaW}}$ up to 10 dB. However, the correlation between enhanced DWR$_{\text{KaW}}$ and MDV is less clear than in the case shown in Kneifel et al. (2015). A more detailed investigation showed that TRIPEx contains only short riming periods of a few minutes duration, while the period analyzed by Kneifel et al. (2015) was considerably longer ($\approx 20$ min). In

general, DWR$_{\text{KaW}}$ is expected to increase for larger particles and strong riming, but detailed sensitivity studies which clearly characterize these dependencies are still missing. Another mode in Figure 12 with larger DWR$_{\text{XKa}}$ of about 3 dB suggests mean particle sizes exceeding 8 mm according to Kneifel et al. (2015). We speculate that this mode might be related to only

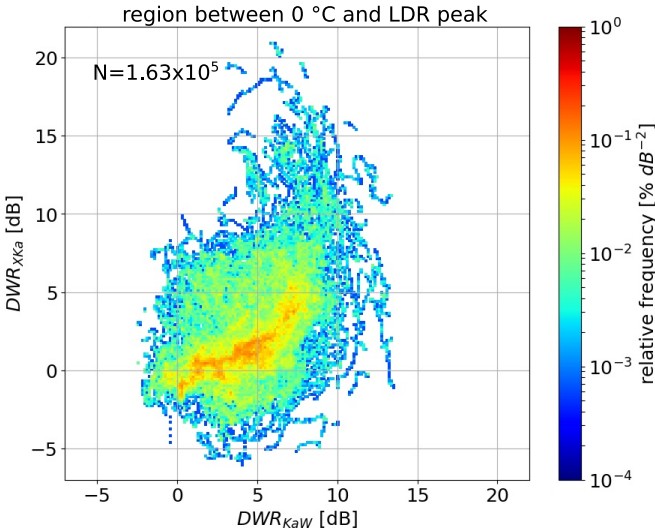

**Figure 11.** Two dimensional histogram of the triple frequency signatures for the region between $0\ ^\circ$C and the LDR maximum in the melting layer normalized by the total number of points N. The color shows the relative frequency and the binning is matching what has been used for Figure 10. Note the log-scale on the colorbar.

slightly rimed aggregates. A larger number of riming events is required to better investigate the sensitivities of MDV and triple-frequency signatures to various degrees of riming, which also would be a very valuable basis to constrain theoretical particle models as developed for example by Leinonen and Szyrmer (2015).

A particularly interesting signature shown in Figure 11 is the very large $DWR_{XKa}$ close to the melting layer. To our knowl-

edge, these features have not yet been described. It is not clear to us whether these signatures are caused by very large aggregates or melting particles. A pure melting of snowflakes should enhance the MDV because of their decrease in size (and thus cross sectional area) as well as drag in the airflow. Early melting can, however, be better detected by LDR: the much larger refractive index of liquid water compared to ice and the initially still asymmetric melting snowflakes result in a much larger depolarization signal as compared to dry snowflakes. Hence, we re-plot Figure 11 to better see the transition from dry snowflakes with

a typical MDV of $1\ \mathrm{ms}^{-1}$ and LDR around -15 dB to larger MDV coinciding with rising LDR as expected for melted snow. Interestingly, the very large $DWR_{XKa}$ show mostly MDV and LDR values associated with unmelted snowflakes. Once the MDV and LDR indicate the onset of melting, the DWRs, especially $DWR_{XKa}$, rapidly decrease. As $DWR_{XKa}$ is strongly related to the mean particle size, the results indicate that the largest snowflake sizes occur before the melting starts. Once snowflakes are completely melted, $DWR_{KaW}$ will be still enhanced due to Mie scattering by the raindrops while $DWR_{XKa}$ will remain close to

0 dB (Tridon et al., 2017). However, our corrections for attenuation within the melting layer are certainly incomplete, thus we leave a deeper analysis of that feature to future studies.

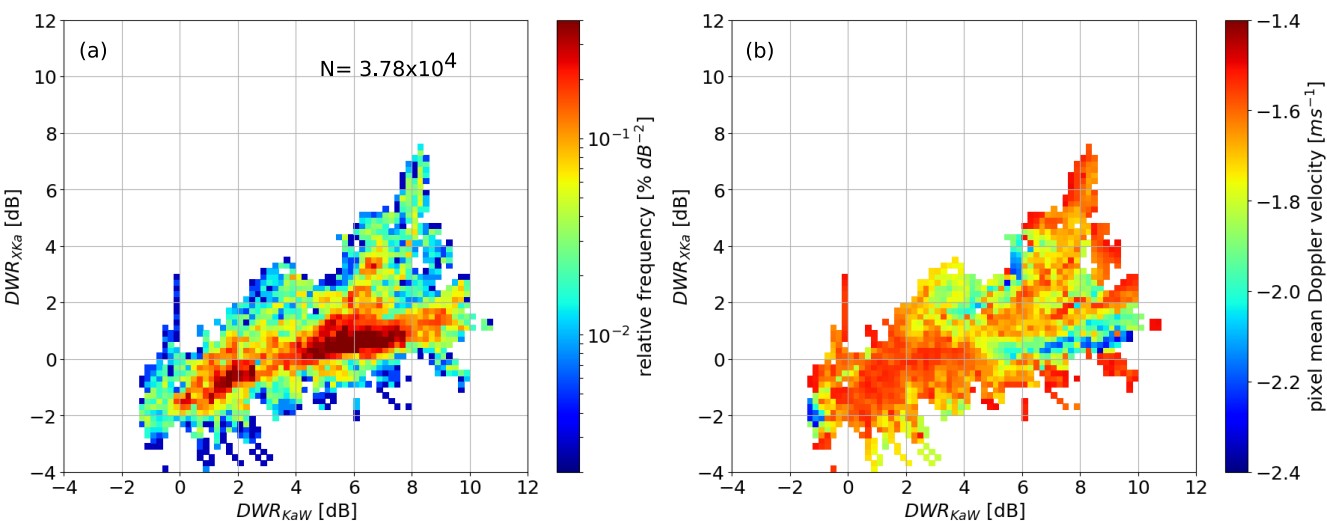

**Figure 12.** Triple frequency signatures for Level 2 data with temperatures between -20 and -1 °C and a mean Doppler velocity (MDV) above 1.5 ms$^{-1}$ in order to select potentially rimed particles. Panel a shows the relative frequency of the observations; Panel b indicates the average MDV of each pixel in the histogram. Note the log-scale on the colorbar in A.

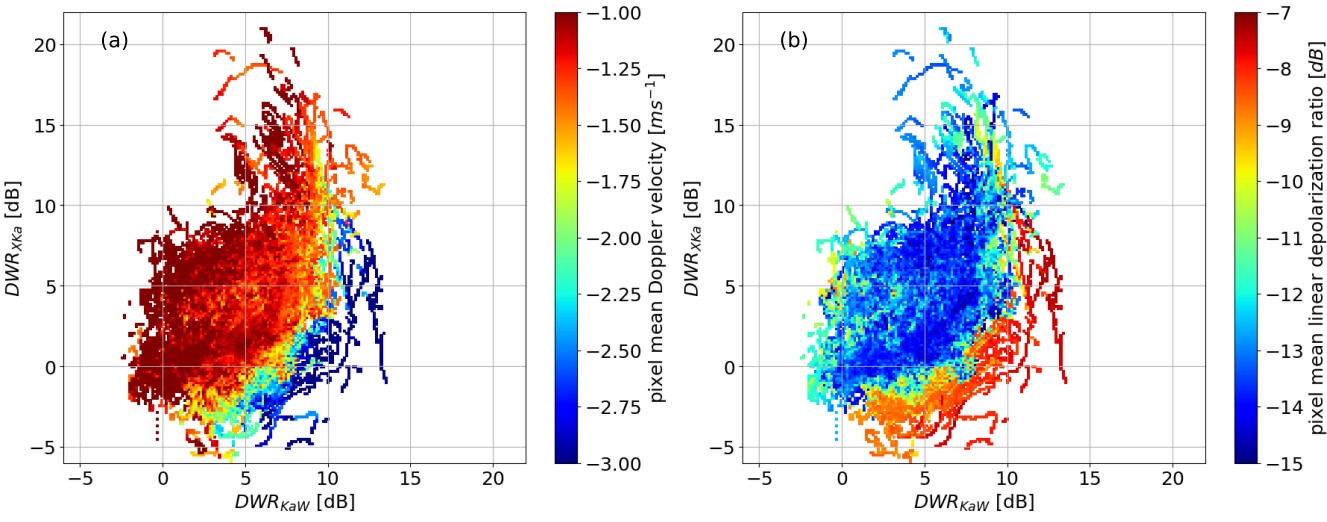

**Figure 13.** Triple-frequency diagrams of observations between 0 °C and the LDR maximum in the melting layer (same as Panel c of Figure 10), but the color in Panel a indicates the average MDV while in Panel b the color shows the average LDR.

## 6 Conclusions

We present the first two-month long dataset of vertically pointing triple-frequency Doppler radar (X, Ka, and W Band) observations of winter clouds at a mid-latitude site (JOYCE-CF, Jülich, Germany). The dataset includes spatio-temporal re-gridded data including offset and attenuation corrections. Several quality flags allow to filter the dataset according to the needs of the specific application. The quality flags have been separated into error and warning flags; we recommend to always apply the error flags, while the warning flags might not be necessary depending on the application. All corrections applied are stored separately in the data files in order to allow the user to recover and work also with data at intermediate processing steps and to potentially apply individual corrections. This might be necessary because the campaign focus was on the ice and snow part of the cloud. Consequently, the correction for path integrated attenuation, might be inappropriate for example for studies investigating the melting layer or rainfall.

The statistical analysis of the ice part of the clouds revealed dominant triple-frequency signatures related to aggregation (hook or bending up feature). In agreement with previous studies, $DWR_{KaW}$ mostly saturates around 7 dB while $DWR_{XKa}$ reaches values of up to 20 dB in regions of presumably intense aggregation close to the melting layer. Due to the large dataset, we were able to investigate the relation between the DWRs and temperature. The first significant increase of aggregation starts around -15 °C where dendritic crystals are known to grow efficiently and favour aggregation. In this zone, $DWR_{KaW}$ mostly increases up to its saturation value of 7 dB. $DWR_{XKa}$ increases mainly below -10 °C. Close to the melting layer, $DWR_{XKa}$ massively increases up to 20 dB, which has not been reported so far. A deeper investigation using LDR and MDV revealed that these extreme $DWR_{XKa}$ are indeed due to large dry aggregates rather than melting particles. Once melting is indicated by larger MDV and LDR, $DWR_{XKa}$ appears to rapidly decrease. Clearly, combined observational and scattering modeling studies are needed to further investigate this transition. Although the dataset contains only a few short riming periods (approximately 1% of the data between -20 and -1 °C), a simple MDV threshold reveals the typical riming signature (flat horizontal line in the triple-frequency space) reported for riming case studies in Kneifel et al. (2015). The statistical analysis of riming is more challenging compared to aggregation. Riming is often connected to larger amounts of super-cooled liquid water, larger vertical air motions, and turbulence, which deteriorate the signal due to liquid water attenuation and enhance effects of imperfect radar volume matching. Riming could be further investigated with this dataset by focusing on single cases, for which it is possible to apply specific corrections and filtering.

The synergy with nearby polarimetric weather radar observations will be investigated in future studies by including the vertical polarimetric profiles matching the JOYCE-CF site based on Quasi-Vertical Profiles (QVPs) (Trömel et al., 2014; Ryzhkov et al., 2016) or Columnar Vertical Profiles (CVPs) (Murphy et al., 2017; Trömel et al., 2018). Also a data release including the W and Ka Band Radar Doppler spectra is planned.

## 7 Data availability

The TRIPEx Level 2 data are available for download at the ZENODO platform (https://doi.org/10.5281/zenodo.1341389). Quicklooks of the TRIPEx dataset are freely accessible via a data quicklook browser (http://gop.meteo.uni-koeln.de/~Hatpro/

dataBrowser/dataBrowser1.html?site=TRIPEX&date=2015-11-20&UpperLeft=3radar_Ze). The raw and Level 1 data and $K_{dp}$ can be requested from the corresponding author.

*Competing interests.* The authors declare that they have no conflict of interest.

*Acknowledgements.* The authors acknowledge the funding provided by the German Research Foundation (DFG) under grant KN 1112/2-1
5    as part of the Emmy-Noether Group OPTIMIce. JD also acknowledges support by the Graduate School of Geosciences of the University of Cologne. We thank the departments S, G and IEK-7 for the technical and administrative support during the field experiment. The majority of data for this dataset were obtained at the JOYCE Core Facility (JOYCE-CF) co-funded by DFG under DFG reserach grant LO 901/7-1. Major instrumentation at the JOYCE site was funded by the Transregional Collaborative Research Center TR32 (Simmer et al., 2015) funded by DFG, and JuXPol by the TERENO (Terrestrial Environmental Observatories) program of the Helmholtz Association (Zacharias et al.,
10    2011). For this work, we used products obtained within the Cloudnet project (part of the EU H2020 project ACTRIS (European Research Infrastructure for the observation of Aerosol, Clouds, and Trace gases)) and developed during the High Definition Clouds and Precipitation for advancing Climate Prediction $HD(CP)^2$ project funded by the German Ministry for Education and Research under grants 01LK1209B and 01LK1502E.

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
