# Peer review of "The TRIple-frequency and Polarimetric radar Experiment for improving process observation of winter precipitation"

_Earth System Science Data, 2018_

## Referee Comment (RC1) · Anonymous Referee #1 · 30 Jan 2019

The TRIple-frequency and Polarimetric radar Experiment for improving process observation of winter precipitation By José Dias Neto et al. The authors present a two-months long dataset of collocated triple-frequency radar observations of clouds and precipitation collected at the Julich Observatory. The main focus of the dataset is on the triple-frequency observations of ice clouds. The dataset is of definite scientific interest and the accompanying article should be published after minor modifications.

Comments: My main suggestion to the authors is to include a table with the list of the events and their description. Since this is a dataset article, such table is definitely needed. The table should also include information such as duration of events, precipi-

tation rate, ground temperature, etc.

Minor comments:

p.4. line 22 "Unlike for the pulsed radars, the JOYRAD-94 range resolution is not fixed, but depends on the gate distance as a consequence of the use of different FMCW chirp settings for different heights."

I would like to suggest that you reword this sentence. A range resolution of a FMCW radar can be selected in a more flexible manner than for a pulsed system, but it does not mean it is not fixed.

p. 7. line 4 "Our relative calibration approach follows the previous triple-frequency study by Kneifel et al. (2015))"

You are using reflectivity matching at the top of ice clouds to mitigate calibration mismatches and differential attenuation in rain, melting layer and ice clouds. Can you elaborate more what kind of impact this approach will have on DWR observations, because this approach implies that the specific attenuation is the same for all range gates. That also means that in some cases the attenuation is overestimated, while in others is underestimated. How important this mismatch for melting layer and ice cloud measurements?

p.16. lines 12-13 "The first triple-frequency signatures from ground-based radars (C, Ka, W band) were presented by Stein et al. (2015) for two case studies."

I guess you meant S-band and not C-band.

---

## Referee Comment (RC2) · Anonymous Referee #2 · 5 Feb 2019

Review of ESSD paper – $2018_142$

This manuscript describes a dataset of two-months of observations from the JOYCE-CF instrument suite in Germany. This work focuses on three ground-based radars at frequencies of 13, 35, and 94 GHz. There is a detailed description of the data processing and filtering methods implemented in producing the triple-frequency co-located observations. These triple-frequency patterns of clouds and precipitation are examined with ancillary meteorological measurements to isolate processes. Aggregation and melting particle signatures are isolated and identified. In general, I believe

this work is unique and important, and builds well on previous triple-frequency radar precipitation studies (i.e., Kulie et al., 2014). Consequently, I think this work should be published after adequately addressing concerns I have outlined below. I would classify my review of this paper as a major amount of minor revisions – however, I want to emphasize that the writing needs to be thoroughly edited and clarified; this is especially important in Section 3, as the details of the filtering of the dataset need to be very clear.

Major Comments:

My main issue with the paper is the writing. Word usage, punctuation issues, and change of author voice (inconsistencies) throughout the manuscript make it a difficult read. The author(s) need to carefully go through the manuscript, as there are several places with incorrect or missing punctuation (i.e., lots of missing commas, semicolons are often used incorrectly). Run on sentences and long phrasing needs to be broken up in places for clarity. Please be careful with tense as well – there are statements voiced as present and past together. And try and avoid passive voice statements if possible.

Some examples of proper comma usage (with comma added):
Page 2, Line 15 . . .characteristic particle size, as well as. . .
Page 3, Line 4 . . .other two radars, which were installed. . .
Page 9, Line 14 . . .close to cloud edges, or for Ze close to. . .

Please be consistent in how the citations are presented (i.e., Löhnert et al., 2015 vs. Löhnert et al. (2015)).

The radars are referred to as X, Ka, W bands and X-band, Ka-band, W-band – please
choose one to use consistently in the manuscript.

It is confusing that throughout the paper the author uses the names of the respective radars (KiXPol, JOYRAD-35, and JOYRAD-94), but in the figures it is often the bands being referenced (ie., X, Ka, W)... it would be best to make these consistent for the reader. I suggest using the names when initially introducing the radars in Section 2, but then use X-, Ka-, and W-band radars in the text when talking about the figures, as it is easier to follow the statements. These inconsistencies are distracting.

In Section 3.2 a Parsivel is used to correct the calibration of the 35GHz radar at the lowest noise-free bin ( 500 m AGL). My question is how confident are the authors that the DSDs at 500 m and the ground are so comparable that they feel this is an ideal means of evaluating the radar? My experience is that the DSD can change a lot within the BL and certainly in 500 m. Additionally, if we assume growth of droplets for these cases, then the shift in the histograms shown in Fig 3 is in the correct direction for a DSD with larger droplets at the surface compared to 500 m. So, my question is how can you separate out changes in DSD as a possibility for the results you see in Fig 3 (since you have no way of truly knowing the DSD at 500 m)? In other words, how are you accounting for the ambiguity of any growth mechanisms in the lowest 500 m?

In general, Section 3 needs attention, as it could be written more clearly. The data processing and filtering methods are complicated, and I think more attention needs to be focused on cultivating a very clear description of issues presented. As it stands, Section 3 tends to go back and forth between concepts. For example, the 3-minute moving window average is introduced on Page 11, Line 4 (in reference to Figure 5), but it is not explained why it is implemented. Later on, in Section 3.6, the 3-minute moving window average is re-introduced and explained as to why it is used. This is one example of many in Section 3 that I feel muddle the message and confuse the

reader. Another issue is that discussion of the Errors and Warnings is disorganized. I think reorganizing and clarifying sections 3.4, 3.4.1, and 3.6 are needed. All of these sections feel somewhat random and the narrative thread is lost.

DGZ is defined as -17.5 to -12.5 C. . . so you could have some possible DG happening in coldest T range. Why not show what is happening even lower temps (Fig 9), why stop at -20C? Is it possible to make figure 10 with cutoffs of <-17C, between -17 and -12, and >-12C to try and to best isolate the DGZ?

Minor Comments (my suggestion and comments after the – ):

Page 1, Line 1 – a two-month dataset
Page 1, Line 1 – "Doppler cloud" – the X-band does not see cloud droplets, so maybe amend the abstract to say cloud and precipitation radar observations
Page 1, Line 2 – capitalize Core Facility
Page 1, Line 19 – remove extra )
Page 2, Line 5/6 – unclear sentence "The microwave. . ."
Page 2, Line 18 – two months of winter
Page 3, Line 6 – capable of measuring
Page 3, Line 7 – (LDR) – when you define an acronym, put ( ) around it
Page 3, Line 15 – "interrupts" is not the correct usage here
Page 4, Line 5 – already defined LDR, no need to repeat
Page 4, Line 9/10 – Passive voice, rewrite with the clauses reversed "JOYRAD was vertically pointing most of the time, as the main. . ."
Page 4, Line 15 – 2nd
Page 4, Line 23 – "consequence of the use. . ." does not make sense
Page 5, Line 1 – "include the exclusion" does not make sense
Page 5, Line 10 – (DWRs)

[Figure]

Page 6, Line 5/6 – why is nearest neighbor italicized with no dash and then not below and with a dash?

Page 6, Line 6 – ensures conservation of

Page 7, Line 2 – relative to each other

Page 7, Line 4 – extra )

Page 7, Line 16 – Citation needed

Page 7, Line 28 – two-way

Page 8, Line 1 – Inter-radar

Page 9, Line 8/9 – Run on sentence. Break up and clarify

Page 9, Line 10 – inter-radar

Page 9, Line 18/19 – "upper, frozen part" of what? I assume the cloud. Also, this sentence is a run on and needs to be clarified

Page 10, Line 28 – "Apparently"?

Page 10, Line 29 – cannot

Page 10, Line 32/33 – "This..." what does This refer to here?

Page 11, Line 5 – "even more accentuates" is not clear

Page 13, Line 11/12 – "kinks in the..." Could you be more descriptive and specific about the feature you are highlighting here? Perhaps add values?

Page 13, Line 12 – "did not allow to monitor" does not make sense

Page 14, Line 5/6/7 – the "slowly descends" is indicated by MDV – so it would help to add a time/height plot of the MDV here to Figure 7. I think this will help clarify what you are trying to highlight

Page 14, Line 8 – Use of chaff with radar implies aircraft deployed material to scramble a signal. Is that the case here – was this actual chaff from an aircraft coordinated with the site? Or is it that the use of "chaff" is to highlight that there is some garbage in radar signal with high LDR and should be ignored?

Page 16, Line 21 – space (and comma) needed between dB and while

Page 16, Line 24 – estimation of

Page 16, Line 25 – "In the following..." following what?

Page 16, Line 29 – dataset
Page 17, Line 10 – inter-radar
Page 17, Line 17 – "are about to decrease" there is no time information in these plots, how do you know they are about to decrease?
Page 18, Line 22 – "CloudNet product" singular or plural?
Page 20, Line 6 – Since the riming periods were so short, how confident are you in these characterizations?

Figure Comments:

The labeling for the panels in the figures is confusing. Convention is that the panel labels (A, B, C. . .) are in the upper-left corner and outside of the plotting area. Please adjust the labeling of the panels so that it is in line with convention and clearer for the reader. See attached figure as an example.

Figure 4 – Note the log-scale on the colorbars in C and F
Figure 5 – Note the log-scale on the colorbars in C and F and I
Figure 6 – Note the log-scale on the colorbar
Figure 7 – Add a panel for mean Doppler Velocities
Figure 9 – It would be more useful to have these two plots with same x-axis limits for comparison
Figure 10 – Note the log-scale on the colorbars in A and B and C
Figure 11 – Note the log-scale on the colorbar
Do Figure 10 and 11 have the same binning for the histograms?
Figure 12 - Note the log-scale on the colorbar in A

---

## Referee Comment (RC3) · Anonymous Referee #3 · 11 Feb 2019

The paper describes a dataset composed of radar observations collected at vertical incidence at three different frequencies (at X, Ka, and W band). The "level 2" data are available through the ZENODO platform in netcdf format, while original data should be obtained through the corresponding author. Many studies on multiple wavelength radar techniques have been published in the recent years to show the potential of such techniques for improving retrievals of clouds and snow properties. Since collocated measurements at multiple frequencies are not common, the publication of the dataset is welcome. Examples presented in section 4 highlight the potential of this dataset. Therefore, I recommend that the manuscript should be accepted after minor but mandatory revision. The language of the manuscript is quite poor, but I am

confident that after a thorough revision by the author, it can be improved. In the following there is a list of comments/recommendations: 1) The manuscript presents a two-month dataset that authors (in the conclusion) consider a "long-term" one. I am not sure whether "long-term" is appropriate or not. However, it is relevant to add a description of what is in the two-month dataset (how many rain events, snow, a description of main events) 2) KiXPol: the X-band radar implements the STAR-mode and LDR measurements are not possible. However, other measurements, such are the copolar correlation coefficients are useful at vertical incidence. Moreover, since the antenna is rotating, also second order moment of dual polarization measurements can be used. I think that, to be considered as completed, KiXPol data set should include reflectivity, dual-pol, and Doppler measurements (of course, if collected during the experiment). The manuscript reports "Using a pulse duration of 0.3 $\mu$s, we set the radial resolution down to 30 m which is close to the resolution of the other radars". Actually, the radial resolution corresponding to 0.3 $\mu$s is 45 meters. 3) JOYRAD-35: As I understood, all the radars are calibrated by the manufacturer. The 35GHz radar, maybe the most popular instrument of this class, comes with a 2dB bias due to receiver losses (as per communication from the manufacturer). Then the verification of the calibration with a disdrometer shows a further 4 dB underestimation. One can preliminarily use a bias determined by the comparison with disdrometer to analyze data, but, after all, the radar should be inspected. Moreover, the sensitivity declared in Table 1 is not the same resulting from Figure 6. 4) JOYRAD-94: note not all the FMCW radars have variable range resolution. 5) "Inter radar calibration": What is presented is not a radar calibration, but it is a method to normalize data from different radar to make them comparable in the ice part of the cloud. Therefore, the title of the section is misleading. Actually, JOYRAD-35 was the only system that underwent a calibration and a calibration problem was highlighted. What about the other systems? I think that working with three different systems, calibrating accurately each of them before a campaign is mandatory.

---

## Referee Comment (RC4) · Anonymous Referee #4 · 12 Feb 2019

General Comment

The manuscript described a multi-frequency radar reflectivity dataset collected at the JuÌĹlich ObservatorY for Cloud Evolution core facility (JOYCECF), Germany from November 2015 to January 2016. The dataset is expected to be useful to analyze multi dual frequency ratio (DFR or DWR) for ice clouds to identify ice particle types and ice growth processes. Analyses using the dataset can give unique insight into cloud microphysics and to be valuable for radar and cloud microphysics communities. The data were well calibrated and quantified. The 'Level2' data described in the manuscript are available on the ZENODO website. The manuscript well described the data processing

and quality. The manuscript can be accepted after some minor revisions, but some more information about the observations about the observation and data processing are needed in the manuscript and should be added to the data.

Specific comments

Manuscript

1. Abstract: A sentence in lines 10-11 of abstract "we find very strong aggregation..." This should be rephrased carefully, because several previous studies suggested heavy aggregation above melting layer without using DWR (e.g., Ryzhkov et al. 1998). Please specify this is based on large DWR, as mentioned in conclusion (line 7 in p.22 DWRXKa massively increases up to extreme values of 20 dB, which has not been reported so far.). The sentence in the abstract should also follow this conclusion.

2. Section 3.1:

- Did you use IQ signals to change average time and range-gate spacing? I think that pulse width, PRF, and the number of pulse average are important information to justify the data quality and understand error sources.

- How many data pointes were used for the nearest neighbor interpolation?

3. Section 3.2:

- What are the variability of simulated attenuation and the uncertainty in scattering calculation assumptions (T-matrix)? Could you describe how much the variability/uncertainty in the scattering calculation assumptions impact the calculated attenuation?

- Does the quality vary with height? Did you consider radar data sampling volume in the quality?

- I suppose that attenuations by ice hydrometeors and supercooled liquid droplets were not considered. I suggest mentioning thin in the manuscript or data files.

4. Figure:

- Figure 4: I could not clearly see differences between Panel A and Panel D, and Panel B and Panel E. I think that zoom-up plots for 15:00-23:00 would be better.

- Figure 5: Same as Fig. 4. I think that zoom-up plots for 13:00-15:00 would be better.

5. Others: - P. 14, line 9: Should be Z_dr.

- What is the LDR limitation value for each radar? - P. 20, line 9: Should be "suggests."

Data 1. Each file has more than 0.5 GB in size. This is not fairly small to promote using the product. Which version of NetCDF was used? I would recommend using NetCDF4, which can save data space very well.

2. Are you willing to employ the CF-Radial format (https://ral.ucar.edu/projects/titan/docs/radial_formats/CfRadialDoc.pdf)? I understand that this format may need more variables that are not important for this observation (e.g., azimuth, elevation). But if you think to extend this triple-frequency observation and the datasets to different scan strategies (e.g., slant angle, RHI) and more general radar communities, the CF-Radial format can offer the capability to extend to different observations.

3. Computed attenuation amount at each range gate is also important data. I recommend including the attenuation by hydrometeors (two-way total attenuation or specific attenuation dB/km) in the data files.

4. The initial analysis in the manuscript used temperature information, which is very important additional information to identify ice particle types and particle growth processes. I think that the temperature data should be included in the data files. 5. Pulse width, actual range resolution, range gate spacing, PRF, the number of integration pulses for each radar are also important parameters. I recommend including the information should in the data files.

6. Have the reflectivity data in files been corrected for the systematic offsets mentioned in section 3?

7. Why the many of X-band echo regions were masked compared to Ka/W-band reflectivites?

8. I briefly took a look at the data on Nov. 23. Why did the X-band reflectivity have noisy signatures at lower altitudes? Those signatures were not found in Ka/W-band reflectivities.

9. Why are offset values variable with time in short time periods?

10. What is a variable "nv" in the data files?

---

## Short Comment (SC1) · 21 Feb 2019

The purpose of these comments are to enhance the original manuscript and are suggestions, not formal requests needed to be met for publication. I recommend the manuscript for publication based on the need for triple-frequency datasets in the planning of future spaceborne missions, which takes years of preemptive planning and novel observations. Please address the main concerns of the other reviewers and consider the following comments.

1) Please consider adding recent work on triple frequency snow retrievals into your introduction and discussion:

[Figure]

Gergely, M., Cooper, S. J., and Garrett, T. J.: Using snowflake surface-area-to-volume ratio to model and interpret snowfall triple-frequency radar signatures, Atmos. Chem. Phys., 17, 12011-12030, https://doi.org/10.5194/acp-17-12011-2017, 2017.

Chase, R. J., Finlon, J. A., Borque, P.,McFarquhar, G. M., Nesbitt, S. W.,Tanelli, S., et al. (2018). Evaluation of triple-frequency radar retrieval of snowfall properties using coincident airborne in situ observations during OLYMPEX. Geophysical Research Letters, 45, 5752–5760. https://doi.org/10.1029/2018GL077997

Leinonen, J., Lebsock, M. D., Tanelli, S., Sy, O. O., Dolan, B., Chase, R. J., Finlon, J. A., von Lerber, A., and Moisseev, D.: Retrieval of snowflake microphysical properties from multifrequency radar observations, Atmos. Meas. Tech., 11, 5471-5488, https://doi.org/10.5194/amt-11-5471-2018, 2018.

Grecu, M., Tian, L., Heymsfield, G., Tokay, A., Olson, W., Heymsfield, A. J. and Bansemer A. (2018). Nonparametric Methodology to Estimate Precipitating Ice from Multiple-Frequency Radar Reflectivity Observations. JAMC, 57, 2605 - 2622

Yin, M., Liu, G., Honeyager, R., & Turk, F. J. (2017). Observed differences of triple-frequency radar signatures between snowflakes in stratiform and convective clouds. Journal of Quantitative Spectroscopy & Radiative Transfer, 193, 13–20. https://doi.org/10.1016/j.jqsrt.2017.02.017

2) Figure 8: The red-green colorbar is difficult to read for those who have a color deficiency. Please consider using a red-blue colorbar (python: 'seismic' or cmocean.cm.balance)

3) Figure 9: Consider normalizing each vertical bin to the maximum frequency in that row rather than plotting the frequency. Also please cite the Yuter and Houze (1995) paper for readers who do not know how to read CFADs. See this reference for guidance: http://olympex.atmos.washington.edu/publications/2018/JGR18_McMurdie-etal_TerrEnhancedPrecip.pdf

---

## Referee Comment (RC5) · Anonymous Referee #5 · 26 Feb 2019

**General Impression**

The manuscript "The TRIple-frequency and Polarimetric radar Experiment for improving process observation of winter precipitation" is a well constructed paper detailing a dataset that will be very valuable to the scientific community. The authors clearly state their methodology for intercalibrating three radars of different wavelengths, and they wrap up the manuscript with a brief analysis of the triple-frequency observations of ice clouds and snow. After some minor editing, this manuscript will be ready for publication.

[Figure]

**Detailed Comments**

Page 6: The units for DWR should be dB not dBZ. The units in the data files, however, are correct.

Page 7, line 25: The Rosenkranz citation is incomplete. This paper only deals with water vapor absorption. The other references, a corrigendum for the water vapor paper [1] and a book chapter that covers the other gasses [2], should also be included.

Page 9, lines 1 and 2: "Mie" should be be replaced with "non-Rayleigh."

In the data files, the units for path-integrated attenuation are listed as dBZ, but they should be dB.

**Technical Corrections**

Page 6, line 6: "...ensures to conserve..." Replace the "to" with "we."

Page 10, line 28: "...populate in areas..." Remove "in."

Page 13, line 13: "dBz" should be "dBZ."

Page 16, line 24: "...allows to estimate..." should be "...allows us to estimate..."

Page 21, line 13: "...we recommend to always apply..." should be "...we always recommend applying..."

**References**

[1] P W Rosenkranz, "Correction to 'water vapor microwave continuum absorption: A comparison of measurements and models'," Radio Sci., vol. 34, pp. 1025, Apr 1999.

[2] P W Rosenkranz, "Absorption of microwaves by atmospheric gasses," in Atmospheric Remote Sensing by Microwave Radiometry, M A Janssen, Ed., pp. 37–90. Wiley, 1993.

---

## Author Comment (AC1) · 27 Mar 2019

**General comment by the authors to the Reviewer #1:**

We thank the reviewer for the time and effort in reading the manuscript and for the constructive comments and suggestion which we in the following address point by point (our answers are marked with blue font color and the modified text are in italic blue).

**Reviewer #1**

The TRIple-frequency and Polarimetric radar Experiment for improving process observation of winter precipitation By José Dias Neto et al. The authors present a two-months long dataset of collocated triple-frequency radar observations of clouds and precipitation collected at the Julich Observatory. The main focus of the dataset is on the triple-frequency observations of ice clouds. The dataset is of definite scientific interest and the accompanying article should be published after minor modifications.

Comments:

1) My main suggestion to the authors is to include a table with the list of the events and their description. Since this is a dataset article, such table is definitely needed. The table should also include information such as duration of events, precipitation rate, ground temperature, etc.

A: Thanks for this very good suggestion that has been also raised by reviewer #3. We included a table in the new section 4 including the information about the weather conditions for each day of measurements. The table contains the following variables: temperature at 2 m (maximum and minimum), precipitation rate (maximum and mean), accumulated precipitation, liquid water path (maximum and mean) and column integrated water path (maximum and mean). This table also contains the duration of the main weather events classified in four categories (non-precipitating ice clouds, stratiform rain, rain showers and shallow mixed-phase clouds).

Minor comments:

2) p.4. line 22 "Unlike for the pulsed radars, the JOYRAD-94 range resolution is not fixed,but depends on the gate distance as a consequence of the use of different FMCW chirp settings for different heights."

I would like to suggest that you reword this sentence. A range resolution of a FMCW radar can be selected in a more flexible manner than for a pulsed system, but it does not mean it is not fixed.

**A: We agree, we have reworded the sentence according to the reviewer suggestion.**

*The FMCW system allows the user to set different range resolutions for different altitude by acting on the frequency modulation settings (chirp sequence).*

3) p. 7. line 4 "Our relative calibration approach follows the previous triple-frequency study by Kneifel et al. (2015))"

You are using reflectivity matching at the top of ice clouds to mitigate calibration mis-matches and differential attenuation in rain, melting layer and ice clouds. Can you elaborate more what kind of impact this approach will have on DWR observations, because this approach implies that the specific attenuation is the same for all rangegates. That also means that in some cases the attenuation is overestimated, while in others is underestimated. How important this mismatch for melting layer and ice cloud measurements?

**A: This is a good point that we had partly covered in our previous section 3.4, and we agree with the reviewer that this approach has an impact on DWR observations. For this reason, we have extended our discussion in section 3.5 of the reviewed manuscript. Specifically, we use the Ze profile from Ka band  as our reference in the ice part of the clouds because of multiple reasons listed in section 3.5 (best sensitivity, relatively calibrated with disdrometer, larger dynamic range at high altitude). Unfortunately, we are not able to correct for the attenuation produced by hydrometeors because we do not have any information of their distribution along the profile. Assuming that attenuation is always larger for the higher frequencies and taking the Ka-band as reference point we have to accept that all the reflectivity values at the top of the cloud might be**

**underestimated. Concerning the reflectivity difference, our approach ensures correct DWRs at the top of the cloud while the computed DWRs result to be increasingly underestimated at lower levels.**

4) p.16. lines 12-13 "The first triple-frequency signatures from ground-based radars (C,Ka, W band) were presented by Stein et al. (2015) for two case studies."I guess you meant S-band and not C-band.

**A: Corrected**

---

## Author Comment (AC2) · 27 Mar 2019

**General comment by the authors to the Reviewer #2:**

**We thank the reviewer for the time and effort in reading the manuscript and for the constructive comments and suggestion which we in the following address point by point (our answers are marked with blue font color and the modified text are in italic blue).**

**Reviewer #2**

This manuscript describes a dataset of two-months of observations from the JOYCE-CF instrument suite in Germany. This work focuses on three ground-based radarsat frequencies of 13, 35, and 94 GHz. There is a detailed description of the data processing and filtering methods implemented in producing the triple-frequency co-located observations. These triple-frequency patterns of clouds and precipitation are examined with ancillary meteorological measurements to isolate processes. Aggregation and melting particle signatures are isolated and identified. In general, I believe this work is unique and important, and builds well on previous triple-frequency radar precipitation studies (i.e., Kulie et al., 2014). Consequently, I think this work should be published after adequately addressing concerns I have outlined below. I would classify my review of this paper as a major amount of minor revisions – however, I want to emphasize that the writing needs to be thoroughly edited and clarified; this is especially important in Section 3, as the details of the filtering of the dataset need to be very clear.

Major Comments:

1) My main issue with the paper is the writing. Word usage, punctuation issues, and change of author voice (inconsistencies) throughout the manuscript make it a difficult read. The author(s) need to carefully go through the manuscript, as there are several places with incorrect or missing punctuation (i.e., lots of missing commas, semicolons are often used incorrectly). Run on sentences and long phrasing needs to be broken up in places for clarity. Please be careful with tense as well – there are statements voiced as present and past together. And try and avoid passive voice statements if possible.

**We thank the reviewer for all the specific comments on this point. Apart from the punctual corrections, the text has undergone an extensive review and we hope that the writing quality has sufficiently improved compared to the first submission.**

Some examples of proper comma usage (with comma added):

2) Page 2, Line 15 . . . characteristic particle size, as well as…

3) Page 3, Line 4 . . . other two radars, which were installed…

4) Page 9, Line 14 . . . close to cloud edges, or for Ze close to…

**A: We thank the reviewer for the examples. We reworded those sentences. We tried our best to apply the reviewer's suggestion throughout the text whenever appropriate.**

5) Please be consistent in how the citations are presented (i.e., Löhnert et al., 2015 vs.Löhnert et al. (2015)).

**A: We reviewed the entire manuscript and made the citation style consistent.**

6) The radars are referred to as X, Ka, W bands and X-band, Ka-band, W-band – please choose one to use consistently in the manuscript. It is confusing that throughout the paper the author uses the names of the respective radars (KiXPol, JOYRAD-35, and JOYRAD-94), but in the figures it is often the bands being referenced (ie., X, Ka, W) . . . it would be best to make these consistent for the reader. I suggest using the names when initially introducing the radars in Section 2,but then use X-, Ka-, and W-band radars in the text when talking about the figures, as it is easier to follow the statements. These inconsistencies are distracting

**A: Thanks for this suggestion, after section 2 the radars are referred  as  X Band, Ka Band, W Band.**

7) In Section 3.2 a Parsivel is used to correct the calibration of the 35GHz radar at the lowest noise-free bin ( 500 m AGL). My question is how confident are the authors that the DSDs at 500 m and the ground are so comparable that they feel this is an ideal means of evaluating the radar? My experience is that the DSD can change a lot with in the BL and certainly in 500 m. Additionally, if we assume growth of droplets for these

cases, then the shift in the histograms shown in Fig 3 is in the correct direction for a DSD with larger droplets at the surface compared to 500 m. So, my question is how can you separate out changes in DSD as a possibility for the results you see in Fig 3 (since you have no way of truly knowing the DSD at 500 m)? In other words, how are you accounting for the ambiguity of any growth mechanisms in the lowest 500 m?

**A: The calibration method using Parsivel as reference assumes that the rainfall properties at 500 m and ground level are similar in a statistical sense. By assuming this, we are already avoiding the discrepancies we would face by directly comparing time series of simulated and observed Z. However, we agree with the reviewer that this method is prone to uncertainties, especially if processes like drop growth, drop breakup, or evaporation systematically alter the rain distribution within the lowest 500m. Nevertheless, we are confident in our approach because we compared the estimated offset to other techniques (for example, applying the reflectivity saturation technique described in Hogan et al., JTECH, 2003 using a W-Band radar). Recently, we were able to apply this method to several radars with smaller blind zones and found very consistent results. We also applied a different test to the rain cases we selected for this study. Based on the Parsivel PSD, we calculate a reflectivity profile assuming constant PSD and accounting for attenuation. Then we compare the measured reflectivity profile to the attenuated Parsivel-Ze profile at the first useable range gates (500-600m). In that way, we are able to see whether the reflectivity change is mainly due to attenuation or whether there might be other processes affecting the Ze profile. In two of the three cases, the measured changes in the profile are very close to the ones predicted by attenuation and constant PSD (shown in figure below). Of course, this does not answer the question how much the PSD is altered below 500m. Our main motivation to analyze several rainfall cases is that it is rather unlikely that the same processes are present in the different rain events in a similar way. Hence, we are aware that the method is prone to uncertainties but the fact, that the variation of estimated offsets is much smaller than the offsets itself indicates that the method has merit. We extended the discussion of the technique in Section 3.3 in order to address the issues raised by the reviewer.**

[Figure]

8) In general, Section 3 needs attention, as it could be written more clearly. The data processing and filtering methods are complicated, and I think more attention needs to be focused on cultivating a very clear description of issues presented. As it stands,Section 3 tends to go back and forth between concepts. For example, the 3-minute moving window average is introduced on Page 11, Line 4 (in reference to Figure 5),but it is not explained why it is implemented. Later on, in Section 3.6, the 3-minute moving window average is re-introduced and explained as to why it is used. This is one example of many in Section 3 that I feel muddle the message and confuse the reader. Another issue is that discussion of the Errors and Warnings is disorganized. I think reorganizing and clarifying sections 3.4, 3.4.1, and 3.6 are needed. All of these sections feel somewhat random and the narrative thread is lost.

**A: We agree and are grateful for the suggestions. We modified the structure of the sections in order to separate concepts and avoid confusion. Now section 3 is only dedicated to the data processing as the section title suggest. The former sections 3.4.1, 3.5 and 3.6 have been moved to a new section 4 titled "overview of the dataset". In this section, we give an example of applying the filtering flags, show remaining limitations of the dataset and present the sensitivity of the X, Ka and W Band radars during the campaign. Moreover, the discussion of error and warning flags has been made more linear and easier to follow (see the new text at sections 4.1 and 4.2).**

9) DGZ is defined as -17.5 to -12.5 C...so you could have some possible DG happening in coldest T range. Why not show what is happening even lower temps (Fig 9), why stop at -20C? Is it possible to make figure 10 with cutoffs of <-17C, between -17 and-12, and >-12C to try and to best isolate the DGZ?

**We modified both axis as suggested, but we limited the y axis to -30 C. The main reason for this is that both DWRs are very small at temperatures lower than -30. A figure covering the entire temperature range has been made available as supplementary material. Regarding Figure 10: We agree that the suggested temperature regimes better cover the DGZ according to previous cloud chamber experiments. Due to the fact that we lack direct measurements of temperature and that the CloudNet temperature is based on model data, we decided to enlarge the suggested temperature range to -20 to -10°C. In this way the DGZ is certainly included and the main feature of increasing DWR is captured as well. We have included a discussion about this point in the manuscript at section 5.1.**

Minor Comments (my suggestion and comments after the – ):

10) Page 1, Line 1 – a two-month dataset
**A: Done**

11) Page 1, Line 1 – "Doppler cloud" – the X-band does not see cloud droplets, so maybe amend the abstract to say cloud and precipitation radar observations

**A: Agree**

12) Page 1, Line 2 – capitalize Core Facility **A: Done**

13) Page 1, Line 19 – remove extra **A: Done**

14) Page 2, Line 5/6 – unclear sentence "The microwave..."
**A: The sentence has been reworded**

15) Page 2, Line 18 – two months of winter
**A: Done**

16) Page 3, Line 6 – capable of measuring
**A: Done**

17) Page 3, Line 7 – (LDR) – when you define an acronym, put ( ) around it
**A: Done**

18) Page 3, Line 15 – "interrupts" is not the correct usage here
**A: The phrase has been reworded**

*After each complete rotation, the radar stops the measurements for few seconds before the next scan starts, introducing thus a small measurement gap at each scan routine.*

19) Page 4, Line 5 – already defined LDR, no need to repeat
**A: Agree**

20) Page 4, Line 9/10 – Passive voice, rewrite with the clauses reversed "JOYRAD was vertically pointing most of the time, as the main . . ."
 **A: Accepted**

21) Page 4, Line 15 – 2nd
**A: Done**

22) Page 4, Line 23 – "consequence of the use . . ." does not make sense
**A: We reworded this sentence**

*The FMCW system allows the user to set different range resolutions for different altitude by acting on the frequency modulation settings (chirp sequence).*

23) Page 5, Line 1 – "include the exclusion" does not make sense
**A: We changed our word choices**

*These processing steps include the detection and removal of measurements affected by ground clutter, an offset correction of the radars based on independent sources, the compensation for estimated differential attenuation caused by atmospheric gases ...*

24) Page 5, Line 10 – (DWRs)
**A: Accepted**

25) Page 6, Line 5/6 – why is nearest neighbor italicized with no dash and then not belowand with a dash?

**A: We avoided the use of italic and made it consistent throughout the text?**

26) Page 6, Line 6 – ensures conservation of
**A: Done**

27) Page 7, Line 2 – relative to each other
**A: Done**

28) Page 7, Line 4 – extra )
**A: Done**

30) Page 7, Line 16 – Citation needed

**We are not sure if we understood the comment properly. We could not identify the sentence where we should add a reference. We just commented on the duration of the selected rain events.**

31) Page 7, Line 28 – two-way
**A: Done**

32) Page 8, Line 1 – Inter-radar
**A: Done**

33) Page 9, Line 8/9 – Run on sentence. Break up and clarify
**A: The sentence has been rephrased according to the reviewer suggestion.**

*The relative offset correction is estimated for each measuring time from the data inside a moving time window of 15 minutes. The selected data are restricted to the reflectivity pairs which are within threshold values defined above.*

34) Page 9, Line 10 – inter-radar
**A: Accepted**

35) Page 9, Line 18/19 – "upper, frozen part" of what? I assume the cloud. Also, this sentence is a run on and needs to be clarified
**A: This phrase has been rewritten**

*The described adjustment technique accounts for all processes that affect relative offsets of the radars in the upper and frozen part of clouds.*

36) Page 10, Line 28 – "Apparently"?
**A: Corrected**

37) Page 10, Line 29 – cannot
**A: Done**

38) Page 10, Line 32/33 – "This . . ." what does This refer to here?
**A: We have avoided implicit subject.**

39) Page 11, Line 5 – "even more accentuates" is not clear
**A: This phrase has been rewritten**

*An additional 3-minutes running-window averaging of the reflectivities keeps the most stable signatures (Panels d and e in Figure 5), further removes scatter, and thus accentuates the aggregation signature in triple-frequency plot ...*

40) Page 13, Line 11/12 – "kinks in the . . ." Could you be more descriptive and specific about the feature you are highlighting here? Perhaps add values?
 **A: We have accepted the reviewer suggestion**

*The X band Ze distribution shows a crisp enhancement of the largest recorded values at 2 km from 30 dBZ to 40 dBZ.*

41) Page 13, Line 12 – "did not allow to monitor" does not make sense
**A: We have clarified our sentence**

*The X Band sensitivity limitations did not allow to observe signals above 7 km with reflectivities below -10 dBZ…*

42) Page 14, Line 5/6/7 – the "slowly descends" is indicated by MDV – so it would help to add a time/height plot of the MDV here to Figure 7. I think this will help clarify what you are trying to highlight

**A: Thanks for your suggestion, but decided to not include the MDV plot. It takes more than 4 hours for the chaff to descend 4 km. In our opinion the plot of MDV does not  give any additional information but it would overload the plot.**

43) Page 14, Line 8 – Use of chaff with radar implies aircraft deployed material to scramble a signal. Is that the case here – was this actual chaff from an aircraft coordinated with the site? Or is it that the use of "chaff" is to highlight that there is some garbage in radar signal with high LDR and should be ignored?

**A: This chaff event was not coordinated with the site. The most probable explanation is that the chaff was deployed by a military aircraft during training operations, and than the metal particles entered the radar observed volume.**

44) Page 16, Line 21 – space (and comma) needed between dB and while
**A: Corrected**

45) Page 16, Line 24 – estimation of
**A: Corrected**

46) Page 16, Line 25 – "In the following . . ." following what?
**A: In the following sections …**

47) Page 16, Line 29 – dataset
**A: Corrected**

49) Page 17, Line 10 – inter-radar
**A: Corrected**

50) Page 17, Line 17 – "are about to decrease" there is no time information in these plots, how do you know they are about to decrease?

**A: Thanks for the comment. Here we inadvertently used an incorrect term. We have rephrased the sentence**

*In the fourth regime between 0 C and the LDR maximum, $DWR_{KaW}$ tends to further increase while $DWR_{XKa}$ remains constant or even decreases.*

51) Page 18, Line 22 – "CloudNet product" singular or plural?
**A: Corrected, it is plural**

52) Page 20, Line 6 – Since the riming periods were so short, how confident are you in these characterizations?

**We agree with the reviewer that the periods are short for a good characterization, and this is the reason why at Page 20, line 10 (now line 16 of page 23) we say**

*"We speculate, that this mode might be related to only slightly rimed aggregates. A larger number of riming events is required to better investigate the sensitivities of MDV and triple-frequency signatures to various degrees of riming, ..."*

Figure Comments:

53) The labeling for the panels in the figures is confusing. Convention is that the panel labels (A, B, C . . .) are in the upper-left corner and outside of the plotting area. Please adjust the labeling of the panels so that it is in line with convention and clearer for the reader. See attached figure as an example.

**A: Thanks for the suggestion, we agree with the reviewer that the labeling were not standardized. We carefully went through the journal guidelines and corrected the labeling. The journal specifies that the label should be with lowercase and within brackets. Regarding the position, it appears to be common looking at recent ESSD articles to have the labeling in the plot area. We decided to follow this style because it does not increase the Figure.**

54) Figure 4 – Note the log-scale on the colorbars in C and F
**A: Done**

55) Figure 5 – Note the log-scale on the colorbars in C and F and I
**A: Done**

56) Figure 6 – Note the log-scale on the colorbar
**A: Done**

57) Figure 7 – Add a panel for mean Doppler Velocities
**A: See our answer to comment 42 where we address this question.**

58) Figure 9 – It would be more useful to have these two plots with same x-axis limits for comparison
**A: Accepted**

59) Figure 10 – Note the log-scale on the colorbars in A and B and C
**A: Done**

60) Figure 11 – Note the log-scale on the colorbar
**A: Done**

61) Do Figure 10 and 11 have the same binning for the histograms?
**A: Yes, figure 10 and 11 have the same binning and now we mention that in the figure caption**

62) Figure 12 - Note the log-scale on the colorbar in A
**A: Done**

---

## Author Comment (AC3) · 27 Mar 2019

**General comment by the authors to the Reviewer #3:**

**We thank the reviewer for the time and effort in reading the manuscript and for the constructive comments and suggestion which we address in the following point by point (our answers are marked with blue font color and the modified text are in italic blue).**

**Reviewer #3**

The paper describes a dataset composed of radar observations collected at vertical incidence at three different frequencies (at X, Ka, and W band). The "level 2" data are available through the ZENODO platform in netcdf format, while original data should be obtained through the corresponding author. Many studies on multiple wavelength radar techniques have been published in the recent years to show the potential of such techniques for improving retrievals of clouds and snow properties. Since collocated measurements at multiple frequencies are not common, the publication of the dataset is welcome. Examples presented in section 4 highlight the potential of this dataset. Therefore, I recommend that the manuscript should be accepted after minor but mandatory revision. The language of the manuscript is quite poor, but I am confident that after a thorough revision by the author, it can be improved. In the following there is a list of comments/recommendations:

1) The manuscript presents a two-month dataset that authors (in the conclusion) consider a "long-term" one. I am not sure whether "long-term" is appropriate or not. However, it is relevant to add a description of what is in the two-month dataset (how many rain events, snow, a description of main events

**A:  A table describing all events was also suggested by reviewer #1 (see reviewer #1 comment 1). We added this table in section 4. Regarding the statement of a "long-term" data set, we replaced it by "two-month long dataset".**

2) KiXPol: the X-band radar implements the STAR-mode and LDR measurements are not possible. However, other measurements, such are the copolar correlation coefficients are useful at vertical incidence. Moreover, since the antenna is rotating, also second order moment of dual polarization measurements can be used. I think that, to be

considered as completed, KiXPol data set should include reflectivity, dual-pol, and Doppler measurements (of course, if collected during the experiment). The manuscript reports "Using a pulse duration of $0.3\mu s$, we set the radial resolution down to 30 m which is close to the resolution of the other radars". Actually, the radial resolution corresponding to $0.3\mu s$ is 45 meters

**A: We follow the reviewer's suggestion to include polarimetric variables measured by the X Band radar. As the Doppler velocity is already included, the new polarimetric variables added to the dataset are Z_dr, Rho_hv, Phi_dp and K_dp. These variables are included as they are, and no filtering or quality control has been applied to them. It is true that $0.3\mu s$ implies a radial resolution of 45 meters. The explanation of this discrepancy is that the radar software applies an oversampling in order to match the range resolution of 30m of the other radars. We have added a short description of the new variables included in the dataset to the text.**

3) JOYRAD-35: As I understood, all the radars are calibrated by the manufacturer. The 35GHz radar, maybe the most popular instrument of this class, comes with a 2dB bias due to receiver losses (as per communication from the manufacturer). Then the verification of the calibration with a disdrometer shows a further 4 dB underestimation. One can preliminarily use as bias determined by the comparison with disdrometer to analyze data, but, after all, the radar should be inspected. Moreover, the sensitivity declared in Table 1 is not the same resulting from Figure 6.

**A: We agree with the reviewer that the offsets found are quite large and unsatisfactory, but they are not at all unusual. The literature shows calibration offset even larger than found for our system (for example see Protat, A., D. Bouniol, E.J. O'Connor, H. Klein Baltink, J. Verlinde, and K. Widener, 2011: CloudSat as a Global Radar Calibrator. *J. Atmos. Oceanic Technol.,* 28, 445–452, https://doi.org/10.1175/2010JTECHA1443.1). During more recent times, we applied different techniques to several radars at JOYCE from different manufacturers and found calibration biases for each radar in the order of 2-5 dB even though some instruments came directly from inspection or were brand new. One problem is that manufacturers usually only calibrate the internal components of the radar (receiver, transmitter, waveguides). Issues resulting e.g. from imperfect antenna shape, radome attenuation etc. are usually not directly measured because in order to do so, an external target is needed. Very recently, the use of natural volume filling targets are intensively investigated as important monitoring tool for the calibration quality: Maahn et al, AMTD, 2019:**

**https://www.atmos-meas-tech-discuss.net/amt-2019-20/).** **Regarding the sensitivity, the values from the table were indeed slightly different from the figure 6. They have been corrected.**

4) JOYRAD-94: note not all the FMCW radars have variable range resolution.

**A: The reviewer #1 pointed out the same issue. We reworded the sentence.**

*The FMCW system allows the user to set different range resolutions for different altitude by acting on the frequency modulation settings (chirp sequence).*

5) "Inter radar calibration": What is presented is not a radar calibration, but it is a method to normalize data from different radar to make them comparable in the ice part of the cloud. Therefore, the title of the section is misleading. Actually, JOYRAD-35 was the only system that underwent a calibration and a calibration problem was highlighted. What about the other systems? I think that working with three different systems, calibrating accurately each of them before a campaign is mandatory.

**A: All the three radar were calibrated by the manufactures before the campaign. Our calibration methods using rainfall to simulate the according reflectivities is applied  to verify the quality of the initial calibration of the Ka band radar. The additional calibration of the other two radars is required because the manufacturers usually only calibrate the internal components of the radar (receiver, transmitter, waveguides). Issues resulting e.g. from imperfect antenna shape, radome attenuation etc. are usually not directly measured because in order to do so, an external target (caliber) is needed. We have decided to change our discussion in this point to avoid confusion about the term "radar calibration". The new section title, "DWR calibration", and the text that follows emphasize the fact that we are not calibrating the radar systems themselves (Ze is indeed underestimated), but we adjust the reflectivity measurements of X band and W band in order to get DWR values that are calibrated with respect to the known Rayleigh regime in the upper part of the cloud.**

---

## Author Comment (AC4) · 27 Mar 2019

**General comment by the authors to Reviewer #4:**

**We thank the reviewer for the time and effort in reading the manuscript and for the constructive comments and suggestion which we in the following address point by point (our answers are marked with blue font color and the modified text are in italic blue).**

**Reviewer #4**

General Comment: The manuscript described a multi-frequency radar reflectivity dataset collected at the Julich ObservatorY for Cloud Evolution core facility (JOYCE-CF), Germany from November 2015 to January 2016. The dataset is expected to be useful to analyze multi dual frequency ratio (DFR or DWR) for ice clouds to identify ice particle types andice growth processes. Analyses using the dataset can give unique insight into cloud microphysics and to be valuable for radar and cloud microphysics communities. The data were well calibrated and quantified. The 'Level2' data described in the manuscript are available on the ZENODO website. The manuscript well described the data processing and quality. The manuscript can be accepted after some minor revisions, but some more information about the observations about the observation and data processing are needed in the manuscript and should be added to the data.

Specific comments

Manuscript

1. Abstract: A sentence in lines 10-11 of abstract "we find very strong aggregation..."This should be rephrased carefully, because several previous studies suggested heavy aggregation above melting layer without using DWR (e.g., Ryzhkov et al. 1998). Please specify this is based on large DWR, as mentioned in conclusion (line 7 in p.22 DWR$_{XKa}$ massively increases up to extreme values of 20 dB, which has not been reported so far.). The sentence in the abstract should also follow this conclusion.

**A: We agree with the reviewer that previous studies suggest the presence of massive aggregation above the melting layer. For this reason, we reworded this sentence as following:**

*The combination of DWR with mean Doppler velocity and linear depolarization ratio enables us to distinguish signatures of rimed particles and melting snowflakes. The riming signatures in the DWR agree well with results found in previous triple-frequency studies. Close to the melting layer, however, we find very large DWR (up to 20 dB), which have not been reported before.*

Section 3.1:

2) Did you use IQ signals to change average time and range-gate spacing? I think that pulse width, PRF, and the number of pulse average are important information to justify the data quality and understand error sources.

**A: Unfortunately we could not store the IQ signals during the campaign, because it would exceed our storage capacity. The measurements are resampled in post-processing based on the data initially processed by the radar software. The information about pulse repetition frequency, number of spectral average and number of FFT are added to tables 1 and 2 in section 2.**

3) How many data points were used for the nearest neighbor interpolation?
**A: Only one point is used, and it is the closest inside of tolerance window.**

Section 3.2:

4) What are the variability of simulated attenuation and the uncertainty in scattering calculation assumptions (T-matrix)? Could you describe how much the variability/uncertainty in the scattering calculation assumptions impact the calculated attenuation?

**A: The T-matrix approach is an analytic solution of the electromagnetic scattering problem and the uncertainties on that might arise only from an inappropriate selection of the drop shape model or water refractive index. Regarding this, we have adopted the state of the art assumptions as described in the manuscript. However, for the studied cases the computed reflectivity is even very close to the Rayleigh approximation (~1 dB, see the following plot). As a consequence, we**

**strongly believe that uncertainties in the scattering calculations can be considered negligible.**

[Figure]

5) Does the quality vary with height? Did you consider radar data sampling volume in the quality?

**A: Not sure whether we understand the reviewer's question correctly, but we did not consider the radar sampling volume directly for the calibration.**

6) I suppose that attenuations by ice hydrometeors and supercooled liquid droplets were not considered. I suggest mentioning thin in the manuscript or data files.

**A: The reviewer is right, we do not directly correct for the attenuation by ice hydrometeors and supercooled liquid droplets because we do not have information about the vertical distribution of ice hydrometeors and liquid droplets. We are discussing this aspect in section 3.4. After the revision process, we have extended the discussion of this point in section 3.5 following also the comments of reviewer #1 point 3.**

Figure:

7) Figure 4: I could not clearly see differences between Panel A and Panel D, and PanelB and Panel E. I think that zoom-up plots for 15:00-23:00 would be better.

8) Figure 5: Same as Fig. 4. I think that zoom-up plots for 13:00-15:00 would be better.

**A: Thanks for the suggestion, but the paper is mainly focused on ice clouds. A zoom-up as suggested would not help us to show the impact of each flag step.**

Others:

9) P. 14, line 9: Should be $Z_{dr}$.
**A: Corrected**

10) What is the LDR limitation value for each radar?

**A: Not sure whether we understand the reviewer's question correctly, but the Ka Band radar is the only one with LDR capabilities and the minimum measured LDR is ~ -38 dB.**

11) P. 20, line 9: Should be "suggests."
**A: Corrected**

Data

12) Each file has more than 0.5 GB in size. This is not fairly small to promote using the product. Which version of NetCDF was used? I would recommend using NetCDF4,which can save data space very well.

**A: Thanks for the recommendation, but we are already using NetCDF4 to compress the data.**

13) Are you willing to employ the CF-Radial format (https://ral.ucar.edu/projects/titan/docs/radial_formats/CfRadialDoc.pdf)? I understand that this format may need more variables that are not important for this observation (e.g., azimuth, elevation). But if you think to extend this triple-frequency observation and the datasets to different scan strategies (e.g., slant angle, RHI) andmore general radar communities, the CF-Radial format can offer the capability to extend to different observations.

**A: Thanks for the interesting suggestions, and we will consider the CF-Radial format on the following releases of the dataset. For this release we are using SAMD Product Standard (Standardized Atmospheric Measurement Data, version 1.0), and this standard is based on CF conventions 1.6.**

14) Computed attenuation amount at each range gate is also important data. I recommend including the attenuation by hydrometeors (two-way total attenuation or specific attenuation dB/km) in the data files.

**A: Thanks for your suggestion, but we are not able to calculate the attenuation by hydrometeors because we do not have information of hydrometeors distribution for each profile.**

15) The initial analysis in the manuscript used temperature information, which is very important additional information to identify ice particle types and particle growth processes. I think that the temperature data should be included in the data files.

**A: We agree with the reviewer that the temperature provides an important additional information. We added the estimated temperature, pressure and relative humidity profiles to the data files.**

16) Pulse width, actual range resolution, range gate spacing, PRF, the number of integration pulses for each radar are also important parameters. I recommend including the information should in the data files.

**A: We included the pulse repetition frequency, number of FFT and number of spectral average as global attribute in the data files.**

17) Have the reflectivity data in files been corrected for the systematic offsets mentioned in section 3?

**A: The reflectivity variable available in the dataset is corrected using all steps described in section 3, and it also includes the systematic offsets.**

18) Why the many of X-band echo regions were masked compared to Ka/W-band reflectivites?

**A: KiXPOL is a mobile weather X-band scanning radar which we used as a vertically pointing radar for our measurement campaign. Because of its technical**

**specification, it has a lower sensitivity compared to the other radars (see figure 8 and table 6 and the discussion in the relative section).**

19) I briefly took a look at the data on Nov. 23. Why did the X-band reflectivity have noisy signatures at lower altitudes? Those signatures were not found in Ka/W-bandreflectivities.

**A: We describe this noise signatures in the section 4.2 of the updated version of the manuscript, and the most likely explanation for it is occurrence of chaff deployed by military airplane.**

20) Why are offset values variable with time in short time periods?

**A: As described in section 3.4, the remaining offset is calculated using a moving window of 15 min. As result of this technique, the remaining offset is mainly dependent of hydrometeors distribution or wet radome. A sharp frontal passage is likely to cause a sudden variation of the differential attenuation between the various radars.**

21) What is a variable "nv" in the data files?

**A: The "nv" variable is a dimension and it describes the number of vertices of each data point as required by the CF-conventions documentation (http://cfconventions.org/cf-conventions/v1.6.0/cf-conventions.html#cell-boundaries). In the TRIPEx dataset, each data point has 4 vertices. The figure below illustrates position of each vertice (V1, V2, V3, V4) considering a data point at time 't' and height 'h'. Note that in this convention t and h are indexes and not values.**

---

## Author Comment (AC5) · 27 Mar 2019

**General comment by the authors to Randy J. Chase:**

**We thank you for your time and effort in reading the manuscript and for the constructive comments and suggestion which we in the following address point by point (our answers are marked with blue font color).**

**Short comment 1**

**Randy J. Chase**

The purpose of these comments are to enhance the original manuscript and are suggestions, not formal requests needed to be met for publication. I recommend the manuscript for publication based on the need for triple-frequency datasets in the planning of future spaceborne missions, which takes years of preemptive planning and novel observations. Please address the main concerns of the other reviewers and consider the following comments

1) Please consider adding recent work on triple frequency snow retrievals into your introduction and discussion:

Gergely, M., Cooper, S. J., and Garrett, T. J.: Using snowflake surface-area-to-volume ratio to model and interpret snowfall triple-frequency radar signatures, Atmos. Chem.Phys., 17, 12011-12030, https://doi.org/10.5194/acp-17-12011-2017, 2017.

Chase, R. J., Finlon, J. A., Borque, P.,McFarquhar, G. M., Nesbitt, S. W.,Tanelli, S.,et al. (2018). Evaluation of triple-frequency radar retrieval of snowfall properties using coincident airborne in situ observations during OLYMPEX. Geophysical Research Letters, 45, 5752–5760. https://doi.org/10.1029/2018GL077997

Leinonen, J., Lebsock, M. D., Tanelli, S., Sy, O. O., Dolan, B., Chase, R. J., Finlon,J. A., von Lerber, A., and Moisseev, D.: Retrieval of snowflake microphysical prop-erties from multifrequency radar observations, Atmos. Meas. Tech., 11, 5471-5488,https://doi.org/10.5194/amt-11-5471-2018, 2018.

Grecu, M., Tian, L., Heymsfield, G., Tokay, A., Olson, W., Heymsfield, A. J. and Banse-mer A. (2018). Nonparametric Methodology to Estimate Precipitating Ice from

Multiple-Frequency Radar Reflectivity Observations. JAMC, 57, 2605 - 2622Yin, M., Liu, G., Honeyager, R., & Turk, F. J. (2017).

Yin, M., Liu, G., Honeyager, R., & Turk, F. J. (2017) Observed differences of triple-frequency radar signatures between snowflakes in stratiform and convective clouds.Journal of Quantitative Spectroscopy & Radiative Transfer, 193, 13–20.https://doi.org/10.1016/j.jqsrt.2017.02.017

**A: Thanks for the additional references, we added those reference in the introduction.**

2) Figure 8: The red-green colorbar is difficult to read for those who have a color deficiency. Please consider using a red-blue colorbar (python: 'seismic' or cmo-cean.cm.balance)
**A: Thanks for the suggestion, we adopted cmo-cean.cm.balance for the colorbar in this figure.**

3) Figure 9: Consider normalizing each vertical bin to the maximum frequency in that row rather than plotting the frequency. Also please cite the Yuter and Houze(1995) paper for readers who do not know how to read CFADs. See this reference for guidance:

**A: Thanks for the suggestion, we included the reference from Yuter and Houze(1995)  in our manuscript. Regarding the normalization, we are already normalising each row following  Yuter and Houze(1995) definitions.**

http://olympex.atmos.washington.edu/publications/2018/JGR18_McMurdie-etal_TerrEnhancedPrecip.pdf

---

## Author Comment (AC6) · 27 Mar 2019

**General comment by the authors to the Reviewer #5:**

**We thank the reviewer for the time and effort in reading the manuscript and for the constructive comments and suggestion which we in the following address point by point (our answers are marked with blue font color).**

**Reviewer #5**

General Impression

The manuscript "The TRIple-frequency and Polarimetric radar Experiment for improving process observation of winter precipitation" is a well constructed paper detailing a dataset that will be very valuable to the scientific community. The authors clearly state their methodology for inter calibrating three radars of different wavelengths, and they wrap up the manuscript with a brief analysis of the triple-frequency observations of ice clouds and snow. After some minor editing, this manuscript will be ready for publication.

Detailed Comments

1) Page 6: The units for DWR should be dB not dBZ. The units in the data files, however, are correct.

**A: On page 6 line 3, we are saying that the units of the Ze (and not DWR) is in dBZ in order to use the DWR formula.**

**"... dual wavelength ratios (DWRs) which is defined for two wavelengths λ1 and λ2 as DWR = Zeλ1 − Zeλ2  where Zeλ is in dBZ."**

2) Page 7, line 25: The Rosenkranz citation is incomplete. This paper only deals with water vapor absorption. The other references, a corrigendum for the water vapor paper[1] and a book chapter that covers the other gasses [2], should also be included.

**A: We are thankful for the reviewer comment and we agree that the citation was incomplete. We included the suggested references.**

3) Page 9, lines 1 and 2: "Mie" should be be replaced with "non-Rayleigh."In the data files, the units for path-integrated attenuation are listed as dBZ, but they should be dB.

**A: We replaced "Mie" by "non-Rayleigh" as suggested by the reviewer, and we also corrected the unit of the path-integrated attenuation.**

Technical Corrections:

Page 6, line 6: "...ensures to conserve..." Replace the "to" with "we."
**A: Corrected**

Page 10, line 28: "...populate in areas..." Remove "in."
**A: Corrected**

Page 13, line 13: "dBz" should be "dBZ."
**A: Corrected**

Page 16, line 24: "...allows to estimate..." should be "...allows us to estimate..."
**A: Corrected**

Page 21, line 13: "...we recommend to always apply..." should be "...we always recommend applying..."
**A: Corrected**

References[1] P W Rosenkranz, "Correction to 'water vapor microwave continuum absorption: A comparison of measurements and models'," Radio Sci., vol. 34, pp. 1025, Apr 1999.
[2] P W Rosenkranz, "Absorption of microwaves by atmospheric gasses," in Atmospheric Remote Sensing by Microwave Radiometry, M A Janssen, Ed., pp. 37–90.Wiley, 1993.

---

## Author Response (AR1)

https://v2.overleaf.com/project/5b6b577ebbc34a466ebb0c7b

[revised manuscript text omitted]

---

## Author Response (AR2)

https://v2.overleaf.com/project/5b6b577ebbc34a466ebb0c7b

[revised manuscript text omitted]
 yyyy.mm.dd | T 2m [C] max / min | RR [mmh$^{-1}$] max / mean | AR [mm] | LWP [kgm$^{-2}$] max / mean | IWV [kgm$^{-2}$] max / mean | IC [h] | SR [h] | RS [h] | MP [h] |
|---|---|---|---|---|---|---|---|---|---|
| 2015.12.13 | 10.08 / 6.18 | 3.09 / 0.37 | 5.50 | 1.07 / 0.38 | 22.73 / 19.10 | 7 | 0 | 0 | 8 |
| 2015.12.14 | 9.24 / 3.36 | 0.03 / 0.03 | 0.02 | 0.17 / 0.08 | 16.00 / 12.95 | 6 | 0 | 0 | 0 |
| 2015.12.15 | 10.3 / 3.89 | 0.39 / 0.16 | 0.16 | 0.57 / 0.15 | 23.55 / 17.51 | 12 | 2 | 3 | 0 |
| 2015.12.16 | 13.04 / 8.90 | 2.49 / 0.39 | 6.02 | — | — | 0 | 10 | 0 | 7 |
| 2015.12.17 | 16.28 / 12.53 | 3.60 / 0.48 | 0.72 | 1.12 / 0.15 | 25.61 / 20.01 | 8 | 0 | 0 | 6 |
| 2015.12.18 | 13.11 / 8.74 | 0.27 / 0.17 | 0.08 | 0.71 / 0.12 | 26.64 / 16.45 | 10 | 0 | 1 | 2 |
| 2015.12.19 | 13.21 / 9.93 | 0.00 / 0.00 | 0.00 | 0.27 / 0.09 | 25.11 / 22.70 | 8 | 0 | 0 | 0 |
| 2015.12.20 | 13.22 / 11.31 | 0.00 / 0.00 | 0.00 | 0.44 / 0.10 | 23.15 / 20.99 | 22 | 1 | 0 | 0 |
| 2015.12.21 | 12.17 / 9.52 | 0.72 / 0.18 | 0.45 | 0.84 / 0.13 | 23.52 / 14.49 | 3 | 3 | 1 | 6 |
| 2015.12.22 | 14.75 / 10.41 | 2.19 / 0.41 | 1.45 | 0.61 / 0.08 | 26.53 / 22.00 | 16 | 2 | 0 | 8 |
| 2015.12.23 | 13.00 / 4.38 | 0.45 / 0.21 | 0.42 | 0.23 / 0.07 | 14.21 / 11.24 | 4 | 0 | 0 | 8 |
| 2015.12.24 | 14.51 / 4.38 | 5.34 / 0.68 | 1.82 | 1.14 / 0.11 | 22.91 / 15.40 | 6 | 0 | 1 | 3 |
| 2015.12.25 | 13.35 / 7.78 | 3.27 / 0.81 | 4.72 | 0.60 / 0.13 | 24.76 / 18.32 | 15 | 8 | 0 | 4 |
| 2015.12.26 | 15.78 / 7.17 | 0.00 / 0.00 | 0.00 | 0.20 / 0.08 | 22.51 / 17.55 | 4 | 0 | 0 | 4 |
| 2015.12.27 | 14.40 / 6.13 | 0.00 / 0.00 | 0.00 | — | 18.71 / 14.20 | 12 | 0 | 0 | 0 |
| 2015.12.28 | 11.07 / 5.12 | 0.00 / 0.00 | 0.00 | — | 9.56 / 8.57 | 11 | 0 | 0 | 0 |
| 2015.12.29 | 11.87 / 4.35 | 0.00 / 0.00 | 0.00 | 0.34 / 0.08 | 19.78 / 13.80 | 2 | 3 | 0 | 0 |
| 2015.12.30 | 9.40 / 3.77 | 0.00 / 0.00 | 0.00 | 0.05 / 0.04 | 17.80 / 10.93 | 3 | 0 | 0 | 0 |
| 2015.12.31 | 10.31 / 3.53 | 0.69 / 0.20 | 0.47 | 1.01 / 0.22 | 24.39 / 11.82 | 4 | 3 | 2 | 0 |
| 2016.01.01 | 8.45 / 3.46 | 0.30 / 0.13 | 0.10 | 0.83 / 0.13 | 15.42 / 9.85 | 13 | 0 | 0 | 6 |
| 2016.01.02 | 5.94 / 4.11 | 2.88 / 0.72 | 4.69 | 0.42 / 0.14 | 17.80 / 12.89 | 6 | 7 | 0 | 8 |
| 2016.01.03 | 8.29 / 4.84 | 1.86 / 0.44 | 2.95 | 0.93 / 0.23 | 19.85 / 14.45 | 6 | 14 | 0 | 4 |
| 2016.01.04 | 7.74 / 3.66 | 3.57 / 0.81 | 7.06 | — | — | 0 | 10 | 0 | 9 |
| Total | | | | | | 377 | 137 | 47 | 222 |

[revised manuscript text omitted]